# Particle-based Variational Inference with Generalized Wasserstein Gradient Flow

**Ziheng Cheng**[*]
School of Mathematical Sciences
Peking University
`alex-czh@stu.pku.edu.cn`

**Shiyue Zhang**[*]
School of Mathematical Sciences
Peking University
`zhangshiyue@stu.pku.edu.cn`

**Longlin Yu**
School of Mathematical Sciences
Peking University
`llyu@pku.edu.cn`

**Cheng Zhang**[†]
School of Mathematical Sciences and Center for Statistical Science
Peking University
`chengzhang@math.pku.edu.cn`

## Abstract

Particle-based variational inference methods (ParVIs) such as Stein variational gradient descent (SVGD) update the particles based on the kernelized Wasserstein gradient flow for the Kullback-Leibler (KL) divergence. However, the design of kernels is often non-trivial and can be restrictive for the flexibility of the method. Recent works show that functional gradient flow approximations with quadratic form regularization terms can improve performance. In this paper, we propose a ParVI framework, called generalized Wasserstein gradient descent (GWG), based on a generalized Wasserstein gradient flow of the KL divergence, which can be viewed as a functional gradient method with a broader class of regularizers induced by convex functions. We show that GWG exhibits strong convergence guarantees. We also provide an adaptive version that automatically chooses Wasserstein metric to accelerate convergence. In experiments, we demonstrate the effectiveness and efficiency of the proposed framework on both simulated and real data problems.

## 1 Introduction

Bayesian inference is an important method in modern machine learning that provides powerful tools for modeling complex data and reasoning under uncertainty. The core of Bayesian inference is to estimate the posterior distribution given the data. As the posterior distribution is intractable in general, various approximation approaches have been developed, of which variational inference and Markov Chain Monte Carlo are two typical examples. By reformulating the inference problem into an optimization problem, variational inference (VI) seeks an approximation within a certain family of distributions that minimizes the Kullback-Leibler (KL) divergence to the posterior (Jordan et al., 1999; Wainwright & Jordan, 2008; Blei et al., 2016). Equipped with efficient optimization algorithms, VI allows fast training and easy scaling to large datasets. However, the construction of approximating

---

[*]Contributed equally to this work.
[†]Corresponding author.

37th Conference on Neural Information Processing Systems (NeurIPS 2023).

distributions can be restrictive which may lead to poor approximation. Markov chain Monte Carlo (MCMC) methods simulate a Markov chain to directly draw samples from the posterior (Duane et al., 1987; Neal, 2011; Welling & Teh, 2011; Chen et al., 2014). While being asymptotically unbiased, MCMC often takes a long time to converge, and it is also difficult to access the convergence.

Recently, particle based variational inference methods (ParVIs) have been proposed that tend to combine the best of both worlds (Liu & Wang, 2016; Chen et al., 2018; Liu et al., 2019; di Langosco et al., 2021; Fan et al., 2022; Alvarez-Melis et al., 2022). In ParVIs, the approximating distribution is represented as a set of particles, which are iteratively updated by minimizing the KL divergence to the posterior. This non-parametric nature significantly improves the flexibility of ParVIs upon classical VIs, and the interaction between particles also makes ParVIs more particle-efficient than MCMCs. The most well-known particle based VI method is Stein Variational Gradient Descent (SVGD) (Liu & Wang, 2016). It updates the particles by simulating the gradient flows of the KL divergence on a certain kernel related distribution space, where the gradient flows have a tractable form (Liu, 2017; Chewi et al., 2020). However, SVGD relies on the choice of an appropriate kernel function whose design is highly non-trivial and hence could limit the flexibility of the method. Moreover, the required computation of the kernel matrix scales quadratically with the number of particles, which makes it costly to use a large number of particles.

Instead of using kernel induced functional gradients, many attempts have been made to expand the function class for gradient flow approximation (Hu et al., 2018; Grathwohl et al., 2020; di Langosco et al., 2021; Dong et al., 2023). By leveraging the more general neural networks as the function class together with more general regularizers, these approaches have shown improved performance over vanilla SVGD while not requiring expensive kernel computation. However, these methods only use quadratic form regularizers where either the Wasserstein gradient flow or its preconditioned variant is recovered.

In this work, we propose a ParVI method based on a general formulation of minimizing movement scheme in Wasserstein space, which corresponds to a generalized Wasserstein gradient flow of KL divergence. Using Legendre-Fenchel transformation, our method can also be viewed as a functional gradient method with a more general class of regularizers which include the previously used quadratic forms as special cases. We provide a theoretical convergence guarantee of ParVIs with neural-net-estimated vector field for generalized Wasserstein gradient flow, which to the best of our knowledge, has not been established yet. Perhaps surprisingly, our results show that assuming reasonably accurate vector field estimates, the iteration complexity of ParVIs matches the traditional Langevin Monte Carlo under weaker assumptions on the target distribution. As an extension, we also propose an algorithm that can adaptively adjust the Wasserstein metric to accelerate convergence. Extensive numerical experiments on both simulated and real data sets are conducted to demonstrate the efficiency of our method over existing ones.

## 2 Background

**Notations.** Throughout this paper, we use $x$ to denote particle samples in $\mathbb{R}^d$. Let $\mathcal{P}(\mathbb{R}^d)$ denote all the probability distributions on $\mathbb{R}^d$ that are absolute continuous with respect to the Lebesgue measure. We do not distinguish a probabilistic measure with its density function. For $x \in \mathbb{R}^d$ and $p > 1$, $\|x\|_p := (|x_1|^p + \cdots + |x_d|^p)^{1/p}$ stands for the $\ell_p$-norm. The Hölder conjugate of $p$ is denoted by $q := p/(p-1)$. Notation $g^*(\cdot)$ denotes the Legendre transform of a convex function $g(\cdot)$ on $\mathbb{R}^d$.

### 2.1 Particle-based Variational Inference

Let $\pi \in \mathcal{P}(\mathbb{R}^d)$ be the target distribution we wish to sample from. We can cast the problem of sampling as an optimization problem: to construct a distribution $\mu^*$ that minimizes the KL divergence

$$\mu^* := \arg\min_{\mu \in \mathcal{P}'} D_{\mathrm{KL}}(\mu \| \pi), \tag{1}$$

where $\mathcal{P}' \subseteq \mathcal{P}(\mathbb{R}^d)$ is the variational family. Particle-based variational inference methods (ParVIs) is a class of VI methods where $\mathcal{P}'$ is represented as a set of particles. Assume the current particle distribution is $\mu$, then it holds that

$$\frac{d}{d\epsilon}\bigg|_{\epsilon=0} D_{\mathrm{KL}}((id + \epsilon v)_{\#}\mu \| \pi) = -\mathbb{E}_\mu \langle \nabla \log \frac{\pi}{\mu}, v \rangle. \tag{2}$$

ParVIs aim to find the optimal vector field $v$ that minimizes (2) in certain function class. For example, SVGD (Liu & Wang, 2016) restricts $v$ in the unit ball of an reproducing kernel Hilbert space (RKHS) which has a closed-form solution by kernel trick. Meanwhile, Hu et al. (2018); Grathwohl et al. (2020); di Langosco et al. (2021); Dong et al. (2023) consider a more general class of functions for $v$, i.e., neural networks, and minimize (2) with some quadratic form regularizers.

## 2.2 Minimizing Movement Scheme in Wasserstein Space

Assume the cost function $c(\cdot, \cdot) : \mathbb{R}^d \times \mathbb{R}^d \to \mathbb{R}$ is continuous and bounded from below. Define the optimal transportation cost between two probabilistic measure $\mu, \nu$ as:

$$W_c(\mu, \nu) := \inf_{\rho \in \Pi(\mu, \nu)} \int c(x, y) d\rho. \tag{3}$$

Specifically, if $c(x, y) = \|x - y\|_p^p$ for some $p > 1$, then we get the $p$-th power of Wasserstein-p distance $W_p(\mu, \nu)$. Jordan et al. (1998) consider a minimizing movement scheme (MMS) under $W_2$ metric. Given the current distribution $\mu_{kh}$, the distribution for next step is determined by

$$\mu_{(k+1)h} := \arg\min_{\mu \in \mathcal{P}_2(\mathbb{R}^d)} D_{\mathrm{KL}}(\mu \| \pi) + \frac{1}{2h} W_2^2(\mu, \mu_{kh}). \tag{4}$$

When the step size $h \to 0$, $\{\mu_{kh}\}_{k \geq 0}$ converges to the solution of the Fokker-Planck equation

$$\partial_t \mu_t + \mathrm{div}(\mu_t \nabla \log \pi) = \Delta \mu_t. \tag{5}$$

Therefore, MMS corresponds to the deterministic dynamics

$$dx_t = v_t dt, \ v_t = \nabla \log \pi - \nabla \log \mu_t, \tag{6}$$

where $\mu_t$ is the law of $x_t$. (6) is also known as the gradient flow of KL divergence under $W_2$ metric, which we refer to as $L_2$-GF (Ambrosio et al., 2005). Note that the Langevin dynamics $dx_t = \nabla \log \pi(x_t) dt + \sqrt{2} dB_t$ ($B_t$ is the Brownian motion) reproduces the same distribution curve $\{\mu_t\}_{t \geq 0}$ and thus also corresponds to the Wasserstein gradient flow (Jordan et al., 1998).

# 3 Proposed Methods

## 3.1 Minimizing Movement Scheme with A General Metric

We start with generalizing the scope of the aforementioned MMS in Section 2.2 which is under $W_2$ metric.

**Definition 3.1** (Young function). A strictly convex function $g$ on $\mathbb{R}^d$ is called Young function if $g(x) = g(-x), g(0) = 0$, and for any fixed $z \in \mathbb{R}^d \backslash \{0\}$, $hg(\frac{z}{h}) \to \infty$, as $h \to 0$.

**Theorem 1.** *Given a continuously differentiable Young function $g$ and step size $h > 0$, define cost function $c_h(x, y) = g(\frac{x - y}{h})h$. Suppose that $\pi, \mu_{kh} \in \mathcal{P}_{c_h}(\mathbb{R}^d) := \{\mu \in \mathcal{P}(\mathbb{R}^d) : \mathbb{E}_\mu[g(\frac{2x}{h})] < \infty\}$. Under some mild conditions of $g$ (see details in Proposition A.1), $\mathcal{P}_{c_h}(\mathbb{R}^d)$ is a Wasserstein space equipped with Wasserstein distance. Consider MMS under transportation cost $W_{c_h}$:*

$$\mu_{(k+1)h} := \arg\min_{\mu \in \mathcal{P}_{c_h}(\mathbb{R}^d)} D_{\mathrm{KL}}(\mu \| \pi) + W_{c_h}(\mu, \mu_{kh}). \tag{7}$$

*Denote the optimal transportation map under $W_{c_h}$ from $\mu_{(k+1)h}$ to $\mu_{kh}$ by $T_k(\cdot)$. Then we have*

$$\frac{T_k(x) - x}{h} = -\nabla g^* \left( \nabla \log \pi(x) - \nabla \log \mu_{(k+1)h}(x) \right). \tag{8}$$

Please refer to Appendix A for full statements and proofs. Informally, $\mu_{(k+1)h} \approx \mu_{kh}$ for small step size $h$ (Santambrogio, 2017). Further note that $\frac{T_k(x) - x}{h}$ is the optimal velocity field associated with the transport from $\mu_{(k+1)h}$ to $\mu_{kh}$ (and not vice versa). If step size $h \to 0$, then following Jordan et al. (1998), we can recover the dynamics in continuous time:

$$dx_t = v_t dt, \ v_t = \nabla g^*(\nabla \log \pi - \nabla \log \mu_t). \tag{9}$$

We call (9) the generalized Wasserstein gradient (GWG) flow. If we set $g(\cdot) = \frac{1}{2}\|\cdot\|_2^2$ or any positive definite quadratic form $g(\cdot) = \frac{1}{2}\|\cdot\|_H^2$, then (9) reduces to $L_2$-GF (6) or its preconditioned version (Dong et al., 2023) respectively.

## 3.2 Faster Descent of KL Divergence

It turns out that we can leverage the general formulation (9) to explore the underlying structure of different probability spaces and further utilize this geometric structure to accelerate sampling. More specifically, we consider $g(\cdot) = \frac{1}{p}\|\cdot\|_p^p$ for some $p > 1$ and then $g^*(\cdot) = \frac{1}{q}\|\cdot\|_q^q$. Note that if the particles move along the vector field $v_t = \nabla g^*(\nabla \log \frac{\pi}{\mu_t})$, then the descent rate of $D_{\mathrm{KL}}(\mu_t\|\pi)$ is

$$\partial_t D_{\mathrm{KL}}(\mu_t\|\pi) = -\mathbb{E}_{\mu_t}\left\|\nabla \log \frac{\pi}{\mu_t}\right\|_q^q. \tag{10}$$

If we choose $q$ such that $\mathbb{E}_{\mu_t}\left\|\nabla \log \frac{\pi}{\mu_t}\right\|_q^q$ is large, then $D_{\mathrm{KL}}(\mu_t\|\pi)$ decreases faster and the sampling process can be accelerated. We use the following example to further illustrate our idea. Please refer to Appendix B for detailed analysis.

**Example 1.** *Let $\pi = \frac{1}{2}\mathcal{N}(-m, 1) + \frac{1}{2}\mathcal{N}(m, 1)$ and $\mu = \frac{3}{4}\mathcal{N}(-m, 1) + \frac{1}{4}\mathcal{N}(m, 1)$. Then for any $m \geq \frac{1}{80}, q \geq 1$, the following holds:*

$$\frac{0.08}{qm}\left(\frac{m}{3}\right)^q \exp\left(-\frac{m^2}{2}\right) \leq \mathbb{E}_\mu\left\|\nabla \log \frac{\pi}{\mu}\right\|_q^q \leq \frac{0.2}{qm}(4m)^q \exp\left(-\frac{m^2}{2}\right). \tag{11}$$

*However, the KL divergence between $\pi$ and $\mu$ is large: $D_{\mathrm{KL}}(\mu\|\pi) \geq \frac{1}{10\sqrt{2}}$.*

Suppose the target distribution is $\pi$ and we run ParVI with current particle distribution $\mu$. We can expect that, if simply using $L_2$ regularization, *i.e.*, $q = 2$, then for very large $m$, the score divergence is small and thus the decay of KL divergence is extremely slow. However, $D_{\mathrm{KL}}(\mu\|\pi)$ is still large, indicating that it would take a long time for the dynamics to converge to the target. But if we set $q$ much larger, then the derivative of KL divergence would get larger and the convergence can be accelerated.

## 3.3 Algorithm

The forward-Euler discretization of the dynamics (9) is

$$x_{(k+1)h} = x_{kh} + \nabla g^*\left(\nabla \log \frac{\pi}{\mu_{kh}}(x_{kh})\right) h. \tag{12}$$

However, since the score of current particle distribution $\mu_{kh}$ is generally unknown, we need a method to efficiently estimate the GWG direction $\nabla g^*(\nabla \log \frac{\pi}{\mu_{kh}})$. Given the distribution of current particles $\mu$, by the definition of convex conjugate, we have

$$\nabla g^*(\nabla \log \frac{\pi}{\mu}) = \arg\max_v \mathbb{E}_\mu[\langle \nabla \log \frac{\pi}{\mu}, v\rangle - g(v)].$$

If we parameterize $v$ as a neural network $f_w$ with $w \in \mathcal{W}$, then we can maximize the following objective with respect to $w$:

$$\begin{aligned}
\mathcal{L}(w) &:= \mathbb{E}_\mu[\langle \nabla \log \frac{\pi}{\mu}, f_w\rangle - g(f_w)] \\
&= \mathbb{E}_\mu[(\nabla \log \pi)^T f_w + \nabla \cdot f_w - g(f_w)]
\end{aligned} \tag{13}$$

---

**Algorithm 1** GWG: Generalized Wasserstein Gradient Flow

---

**Require:** Unnormalized target distribution $\pi$, initial particles $\{x_0^i\}_{i=1}^n$, initial parameter $w_0$, iteration number $N, N'$, particle step size $h$, parameter step size $\eta$
    **for** $k = 0, \cdots, N-1$ **do**
        Assign $w_k^0 = w_k$
        **for** $t = 0, \cdots, N'-1$ **do**
            Compute

$$\widehat{\mathcal{L}}(w) = \frac{1}{n} \sum_{i=1}^n \nabla \log \pi(x_k^i)^T f_w(x_k^i) + \nabla \cdot f_w(x_k^i) - g(f_w(x_k^i)) \tag{14}$$

            Update $w_k^{t+1} = w_k^t + \eta \nabla_w \widehat{\mathcal{L}}(w_k^t)$
        **end for**
        Update $w_{k+1} = w_k^{N'}$
        Update particles $x_{k+1}^i = x_k^i + h f_{w_{k+1}}(x_k^i)$ for $i = 1, \cdots, n$
    **end for**
    **return** Particles $\{x_N^i\}_{i=1}^n$

---

Here the second equation is by *Stein's identity* (we assume $\mu$ vanishes at infinity). This way, the gradient of $\mathcal{L}(w)$ can be estimated via Monte Carlo methods given the current particles. We summarize the procedures in Algorithm 1.

The exact computation of the divergence term $\nabla_x \cdot f_w(x)$ needs $\mathcal{O}(d)$ times back-propagation, where $d$ is the dimension of $x$. In order to reduce computation cost, we refer to Hutchinson's estimator (Hutchinson, 1989), *i.e.*,

$$\frac{1}{n} \sum_{i=1}^n \nabla \cdot f_w(x_k^i) \approx \frac{1}{n} \sum_{i=1}^n \xi_i^T \nabla(f_w(x_k^i) \cdot \xi_i), \tag{15}$$

where $\xi_i \in \mathbb{R}^d$ are independent random vectors satisfying $\mathbb{E}\xi_i\xi_i^T = I_d$. This is still an unbiased estimator but only needs $\mathcal{O}(1)$ times back-propagation.

## 4 Convergence Analysis without Isoperimetry

In this section, we state our main theoretical results of Algorithm 1. Consider the discrete dynamics:

$$X_{(k+1)h} = X_{kh} + v_k(X_{kh})h, \tag{16}$$

where $v_k$ is the neural-net-estimated GWG at time $kh$. Define the interpolation process

$$X_t = X_{kh} + (t - kh)v_k(X_{kh}), \text{ for } t \in [kh, (k+1)h], \tag{17}$$

and let $\mu_t$ denote the law of $X_t$. Note that here we do not assume isoperimetry of target distribution $\pi$ (*e.g.*, *log-Sobolev inequality*) and hence establish the convergence of dynamics in terms of score divergence, following the framework of non-log-concave sampling (Balasubramanian et al., 2022).

We first make some basic assumptions. For simplicity, only two types of Young function $g^*$ are considered here, which are also the most common choices.

**Assumption 1.** $g^*(\cdot) = \dfrac{1}{q}\|\cdot\|_q^q$ *for some* $q > 1$. *And for any* $k$, $\mathbb{E}_{\mu_{kh}} \left\| v_k - \nabla g^*(\nabla \log \dfrac{\pi}{\mu_{kh}}) \right\|_p^p \leq \varepsilon_k$.

**Assumption 2.** $g^*(\cdot)$ *is* $\alpha$*-strongly convex and* $\beta$*-smooth. Define* $\kappa := \dfrac{\beta}{\alpha}$. *And for any* $k$, $\mathbb{E}_{\mu_{kh}} \left\| v_k - \nabla g^*(\nabla \log \dfrac{\pi}{\mu_{kh}}) \right\|_2^2 \leq \varepsilon_k$.

The two assumptions above ensure the estimation accuracy of neural nets. Note that the preconditioned quadratic form in Dong et al. (2023) is included in Assumption 2. Although the estimation error is not exactly the training objective used in Algorithm 1,the following proposition shows the equivalence between them in some sense.

**Proposition 2.** *Suppose $g(\cdot) = \frac{1}{p}\|\cdot\|_p^p$ for some $p > 1$. Given current particle distribution $\mu$, we can define the training loss $\mathcal{L}_{train}(v) := \mathbb{E}_\mu[\langle \nabla \log \frac{\pi}{\mu}, v \rangle - g(v)]$. The maximizer is $v^* = \nabla g^*(\nabla \log \frac{\pi}{\mu})$ and the maximum value is $\mathcal{L}_{train}^* := \mathcal{L}_{train}(v^*) < \infty$. For any arbitrarily small $\varepsilon_1 > 0$, there exists $\varepsilon_2 := \varepsilon_2(\varepsilon_1, p) < \infty$, such that*

$$\mathbb{E}_\mu \left\| v - \nabla g^*(\nabla \log \frac{\pi}{\mu}) \right\|_p^p \leq \varepsilon_1 \mathcal{L}_{train}^* + \varepsilon_2 [\mathcal{L}_{train}^* - \mathcal{L}_{train}(v)].$$

*Besides, if $p \geq 2$, $\varepsilon_1$ can be 0 while $\varepsilon_2$ is still finite.*

Similar results also hold if $g$ satisfies Assumption 2 since it is equivalent to the case when $p = 2$. Additionally, we expect some properties of the estimated vector fields.

**Assumption 3** (Smoothness of neural nets)**.** *For any $k$, $v_k(\cdot)$ is twice differentiable. For any $p > 1$,*
$$G_p := \sup_{x,y} \lim \frac{\|v_k(x) - v_k(y)\|_p}{\|x - y\|_p} < \infty, \ M_p := \sup_{x,z} \lim_{\delta \to 0^+} \frac{\|\nabla v_k(x + \delta z) - \nabla v_k(x)\|_{op}}{\delta \|z\|_p} < \infty.$$

Note that here we do not assume the smoothness of potential $\log \pi$ explicitly. But informally, $G_p$ and $M_p$ correspond to the Lipschitz constant of the gradient and the Hessian of $\log \pi$, respectively.

Let $\bar{\mu}_{Nh} := \frac{1}{Nh} \int_0^{Nh} \mu_t dt$ and $K_0 := D_{KL}(\mu_0 \| \pi)$. Now we present our main results.

**Theorem 3** (Full version see Theorem D.9)**.** *Under Assumption 1, 3, the following bound holds with proper step size $h$:*

$$\mathbb{E}_{\bar{\mu}_{Nh}} \left\| \nabla \log \frac{\pi}{\bar{\mu}_{Nh}} \right\|_q^q = \tilde{\mathcal{O}} \left( (\frac{M_p K_0 d}{N})^{\frac{q}{q+1}} + \frac{G_2 K_0 d}{N} + \frac{\sum_{k=0}^{N-1} \varepsilon_k}{N} \right). \tag{18}$$

*Here $\tilde{\mathcal{O}}(\cdot)$ hides all the constant factors that only depend on $q$.*

**Theorem 4** (Full version see Theorem D.10)**.** *Under Assumption 2, 3 with $\alpha = 1$, the following bound holds with proper step size $h$:*

$$\mathbb{E}_{\bar{\mu}_{Nh}} \left\| \nabla \log \frac{\pi}{\bar{\mu}_{Nh}} \right\|_2^2 = \mathcal{O} \left( (\frac{\kappa M_2 K_0 d}{N})^{\frac{2}{3}} + \frac{G_2 K_0 (d + \kappa)}{N} + \frac{\sum_{k=0}^{N-1} \varepsilon_k}{N} \right). \tag{19}$$

The proofs in this section are deferred to Appendix D. To interpret our results, suppose $g^*(\cdot) = \frac{1}{q}\|\cdot\|_q^q$ and $\epsilon \lesssim (\frac{M_p}{G_2})^q$. If the neural net $v_k(\cdot)$ can approximate $\nabla g^*(\nabla \log \frac{\pi}{\mu_{kh}})$ accurately (*i.e.*, $\varepsilon_k \lesssim \epsilon$), then to obtain a probabilistic measure $\mu$ such that $\mathbb{E}_\mu \left\| \nabla \log \frac{\pi}{\mu} \right\|_q^q \lesssim \epsilon$, the iteration complexity is $\tilde{\mathcal{O}}(M_p K_0 d \epsilon^{-(1+\frac{1}{q})})$. If we further let $q = p = 2$, the complexity is $\mathcal{O}(K_0 d \epsilon^{-\frac{3}{2}})$, which matches the complexity of Langevin Monte Carlo (LMC) under the Hessian smoothness and the growth order assumption (Balasubramanian et al., 2022). However, noticing that Assumption 3 is similar to Hessian smoothness informally, we can obtain this rate without additional assumption on target distribution. This suggests the potential benefits of particle-based methods.

In addition, our formulation allows a wider range of choices of Young function, including $\|\cdot\|_p^p$ and the preconditioned quadratic form (Dong et al., 2023). This provides wider options of convergence metrics. We refer the readers to Appendix D.4 for more discussions.

## 5   Extensions: Adaptive Generalized Wasserstein Gradient Flow

The GWG framework also allows adaption of the Young function $g$, instead of a fixed one. Similar ideas are also presented in Wang et al. (2018). In this section, we consider a special Young function class $\left\{\frac{1}{p}\|\cdot\|_p^p : p > 1\right\}$ and propose a procedure that adaptively chooses $p$ to accelerate convergence.

**Algorithm 2** Ada-GWG: Adaptive Generalized Wasserstein Gradient Flow

---

**Require:** unnormalized target distribution $\pi$, initial particles $\{x_0^i\}_{i=1}^n$, initial parameter $w_0$, iteration number $N, N'$, step size $h, \eta, \tilde{\eta}$, lower and upper bounds on $p$: $lb, ub$
  **for** $k = 0, \cdots, N-1$ **do**
    Assign $w_k^0 = w_k$
    **for** $t = 0, \cdots, N'-1$ **do**
      Compute

$$\widehat{\mathcal{L}}(w) = \frac{1}{n}\sum_{i=1}^n \nabla\log\pi(x_k^i)^T f_w(x_k^i) + \nabla\cdot f_w(x_k^i) - \frac{1}{p_k}\|f_w(x_k^i)\|_{p_k}^{p_k} \qquad (22)$$

      Update $w_k^{t+1} = w_k^t + \eta\nabla_w\widehat{\mathcal{L}}(w_k^t)$
    **end for**
    Update $w_{k+1} = w_k^{N'}$
    Compute $\widehat{A}(p_k) = \frac{1}{n}\sum_{i=1}^n \frac{1}{p_k}\|f_{w_{k+1}}(x_k^i)\|_{p_k}^{p_k}$
    Update $p_{k+1} = \mathbf{clip}(p_k + \tilde{\eta}\nabla\widehat{A}(p_k), lb, ub)$
    Update particles $x_{k+1}^i = x_k^i + hf_{w_{k+1}}(x_k^i)$ for $i = 1, \cdots, n$
  **end for**
  **return** Particles $\{x_N^i\}_{i=1}^n$

---

Consider the continuous time dynamics $dx_t = f_t(x_t)dt$ and denote the distribution of particles at time $t$ as $\mu_t$, we have the following proposition.

**Proposition 5.** *For $g(\cdot) = \frac{1}{p}\|\cdot\|_p^p$, the derivative of KL divergence has an upper bound:*

$$\partial_t D_{\mathrm{KL}}(\mu_t\|\pi) \le -\frac{1}{p}\mathbb{E}_{\mu_t}\left\|\nabla\log\frac{\pi}{\mu_t}\right\|_q^q + \frac{1}{p}\mathbb{E}_{\mu_t}\left\|\nabla g^*(\nabla\log\frac{\pi}{\mu_t}) - f_t\right\|_p^p. \qquad (20)$$

The proof is in Appendix E. If the neural network $f_t$ can approximate the objective well, *i.e.*, $f_t \approx \nabla g^*(\nabla\log\frac{\pi}{\mu_t})$, then informally we can omit the second term and thus

$$\partial_t D_{\mathrm{KL}}(\mu_t\|\pi) \lesssim -\frac{1}{p}\mathbb{E}_{\mu_t}\left\|\nabla\log\frac{\pi}{\mu_t}\right\|_q^q \approx -\frac{1}{p}\mathbb{E}_{\mu_t}\|f_t\|_p^p =: -A(p). \qquad (21)$$

In order to let KL divergence decrease faster, we can choose $p$ such that $A(p)$ is larger. This leads to a simple adaptive procedure that updates $p$ by gradient ascent *w.r.t.* $A(p)$. In practice, the adjustment of $p$ is delicate and would cause numerical instability if $p$ becomes excessively small or large. Therefore it is necessary to clip $p$ within a reasonable range. We call this adaptive version of GWG, Ada-GWG. The whole training procedure of Ada-GWG is shown in Algorithm 2. Note that (22) can be replaced with Hutchinson's estimator (15) to improve computational efficiency as before.

## 6 Numerical Experiments

In this section, we compare GWG and Ada-GWG with other ParVI methods including SVGD (Liu & Wang, 2016), $L_2$-GF (di Langosco et al., 2021) and PFG (Dong et al., 2023) on both synthetic and real data problems. In BNN experiments, we also test stochastic gradient Langevin dynamics (SGLD). For Ada-GWG, the exponent $p$ is clipped between $1.1$ and $4.0$ unless otherwise specified. Throughout this section, we choose $f_w$ to be a neural network with $2$ hidden layers and the initial particle distribution is $\mathcal{N}(0, I_d)$. We refer the readers to Appendix F for more detailed setups of our experiments. The code is available at `https://github.com/Alexczh1/GWG`.

### 6.1 Gaussian Mixture

Our first example is on a multi-mode Gaussian mixture distribution. Following Dong et al. (2023), we consider the 10-cluster Gaussian mixture where the variances of the mixture components are

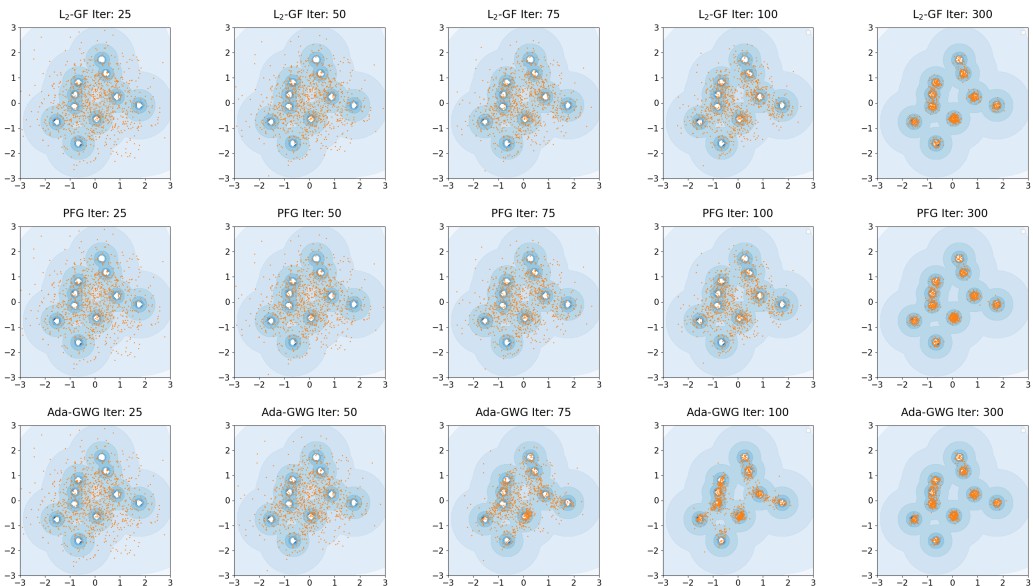

Figure 1: Comparison of sampled particles at different numbers of iterations. **Upper**: $L_2$-GF. **Middle**: PFG. **Lower**: Ada-GWG with $p_0 = 2$.

all 0.1. The number of particles is 1000. Figure 1 shows the scatter plots of the sampled particles at different numbers of iterations. We see that on this simple toy example, PFG performs similarly to the standard $L_2$-GF which does not involve the preconditioner, while Ada-GWG with the initial $p_0 = 2$ significantly accelerates the convergence compared to these two baseline methods. Please refer to appendix for further quantitative comparisons.

### 6.2 Monomial Gamma

To illustrate the effectiveness and efficiency of the adaptive method compared to the non-adaptive counterparts, we consider the heavy tailed Monomial Gamma distribution where the target $\pi \propto \exp(-0.3(|x_1|^{0.9} + |x_2|^{0.9}))$.

We test GWG and Ada-GWG with different choices of the initial values of $p$. The number of particles is 1000. Figure 2 demonstrates the KL divergence of different methods against the number of iterations. The dotted line represents GWG with fixed $p$, while the solid line represents the corresponding Ada-GWG that starts from the same $p$ at initialization.

We see that the adaptive method outperforms the non-adaptive counterpart consistently. Moreover, Ada-GWG can automatically learn the appropriate value of $p$ especially when the initial values of $p$ is set inappropriately. For example, in our case, a relatively small value of $p = 1.5$ would be inappropriate (the dotted green line) for GWG, while Ada-GWG with the same initial value of $p = 1.5$ is able to provide much better approximation by automatically adjusting $p$ during runtime. Consequently, Ada-GWG can exhibit greater robustness when determining the initial value of $p$.

### 6.3 Conditioned Diffusion

The conditioned diffusion example is a high-dimensional model arising from a Langevin SDE, with state $u : [0, 1] \longrightarrow \mathbb{R}$ and dynamics given by

$$du_t = \frac{10u(1 - u^2)}{1 + u^2}dt + dx_t, \quad u_0 = 0, \tag{23}$$

where $x = (x_t)_{t \geq 0}$ is a standard Brownian motion.

This system is commonly used in molecular dynamics to represent the motion of a particle with negligible mass trapped in an energy potential with thermal fluctuations represented by the Brownian forcing Detommaso et al. (2018); Cui et al. (2016). Given the perturbed observations $y$, the goal

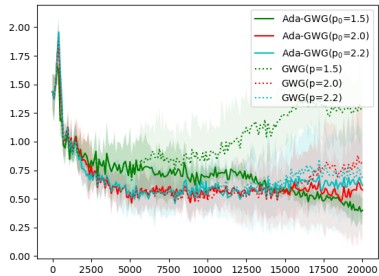 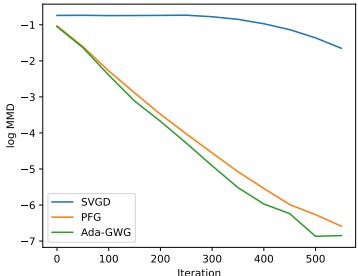

Figure 2: KL divergence of different methods. Solid line: Ada-GWG. Dotted line: GWG counterpart.

Figure 3: Comparison among PFG, Ada-GWG, SVGD in conditioned diffusion example.

is to infer the posterior of the driving process $p(x|y)$. The forward operator is defined by $\mathcal{F}(x) = (u_{t_1}, \cdots, u_{t_{20}}) \in \mathbb{R}^{20}$, where $t_i = 0.05i$. This is achieved by discretizing the above SDE (23) using an Euler-Maruyama scheme with step size $\Delta t = 0.01$; therefore the dimensionality of the problem is 100. The noisy observations are obtained as $y = \mathcal{F}(x_{\text{true}}) + \xi \in \mathbb{R}^{20}$, where $x_{\text{true}}$ is a Brownian motion path and $\xi \sim \mathcal{N}(0, \sigma^2 I)$ with $\sigma = 0.1$. The prior is given by the Brownian motion $x = (x_t)_{t \geq 0}$.

We test three algorithms: PFG, Ada-GWG, and SVGD, with $n = 1000$ particles. To obtain the ground truth posterior, we run LMC with 1000 particles in parallel, using a small step size $h = 10^{-4}$ for 10000 iterations. Figure 3 reports the logarithmic Maximum Mean Discrepancy (MMD) curves against iterations. We observe that Ada-GWG provides best performance compared to the other methods.

### 6.4 Bayesian Neural Networks

We compare our algorithm with SGLD and SVGD variants on Bayesian neural networks (BNN). Following Liu & Wang (2016), we conduct the two-layer network with 50 hidden units and ReLU activation function, and we use a $\mathrm{Gamma}(1, 0.1)$ prior for the inverse covariances. The datasets are all randomly partitioned into 90% for training and 10% for testing. The mini-batch size is 100 except for Concrete on which we use 400. The particle size is 100 and the results are averaged over 10 random trials. Table 1 shows the average test RMSE and NLL and their standard deviation. We see that Ada-GWG can achieve comparable or better results than the other methods. And the adaptive method consistently improves over $L_2$-GF. Figure 4 shows the test RMSE against iterations of different methods on the Boston dataset. We can see that for this specific task, setting $p = 3$ produces better results than when $p = 2$. Although $L_2$-GF (i.e., GWG with $p = 2$) is sub-optimal, our adaptive method (i.e., Ada-GWG with $p_0 = 2$) makes significant improvements and demonstrates comparable performance to the optimal choice of $p = 3$. This suggests that our adaptive method is robust even if the initial exponent choice is not ideal. More comparisons of convergence results and hyperparameter tuning details can be found in the appendix.

Table 1: Averaged test RMSE and test negative log-likelihood of Bayesian Neural Networks on several UCI datasets. The results are averaged from 10 independent runs.

| | Avg. Test RMSE | | | | Avg. Test NLL | | | |
|---|---|---|---|---|---|---|---|---|
| **Dataset** | **SGLD** | **SVGD** | $L_2$**-GF** | **Ada-GWG** | **SGLD** | **SVGD** | $L_2$**-GF** | **Ada-GWG** |
| BOSTON | $3.011_{\pm 0.15}$ | $2.774_{\pm 0.08}$ | $3.072_{\pm 0.10}$ | $\mathbf{2.721}_{\pm 0.08}$ | $2.496_{\pm 0.03}$ | $2.444_{\pm 0.02}$ | $2.547_{\pm 0.14}$ | $\mathbf{2.434}_{\pm 0.02}$ |
| CONCRETE | $5.583_{\pm 0.25}$ | $4.436_{\pm 0.08}$ | $4.343_{\pm 0.11}$ | $\mathbf{3.871}_{\pm 0.10}$ | $3.184_{\pm 0.04}$ | $3.035_{\pm 0.02}$ | $3.053_{\pm 0.03}$ | $\mathbf{2.826}_{\pm 0.02}$ |
| POWER | $4.089_{\pm 0.11}$ | $3.972_{\pm 0.02}$ | $4.014_{\pm 0.02}$ | $\mathbf{3.944}_{\pm 0.01}$ | $2.840_{\pm 0.02}$ | $2.809_{\pm 0.01}$ | $2.824_{\pm 0.01}$ | $\mathbf{2.802}_{\pm 0.01}$ |
| WINEWHITE | $0.677_{\pm 0.01}$ | $0.664_{\pm 0.01}$ | $0.666_{\pm 0.01}$ | $\mathbf{0.660}_{\pm 0.01}$ | $1.033_{\pm 0.01}$ | $1.014_{\pm 0.01}$ | $1.015_{\pm 0.01}$ | $\mathbf{1.006}_{\pm 0.01}$ |
| WINERED | $0.600_{\pm 0.01}$ | $0.579_{\pm 0.01}$ | $0.581_{\pm 0.01}$ | $\mathbf{0.575}_{\pm 0.01}$ | $0.910_{\pm 0.01}$ | $0.887_{\pm 0.02}$ | $0.860_{\pm 0.02}$ | $\mathbf{0.839}_{\pm 0.02}$ |
| PROTEIN | $\mathbf{4.560}_{\pm 0.04}$ | $4.779_{\pm 0.03}$ | $4.867_{\pm 0.01}$ | $4.686_{\pm 0.02}$ | $\mathbf{2.934}_{\pm 0.01}$ | $2.984_{\pm 0.01}$ | $3.003_{\pm 0.00}$ | $2.964_{\pm 0.00}$ |

## 7 Conclusion

We introduced a new ParVI method, called GWG, which corresponds to a generalized Wasserstein gradient flow of KL divergence. We show that our method has strong convergence guarantees in discrete time setting. We also propose an adaptive version, called Ada-GWG, that can automatically

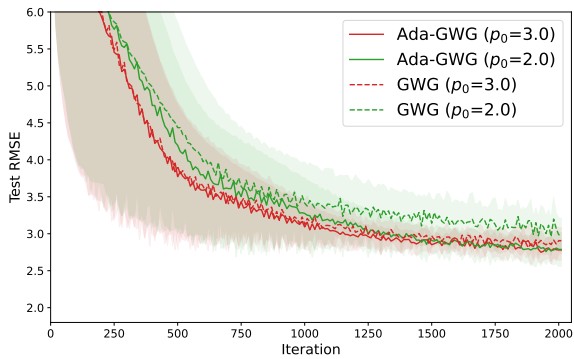

Figure 4: Test RMSE for the Bayesian Neural Networks on Boston dataset. The number in parentheses specifies the initial exponent $p_0$. The results are averaged from 10 independent runs.

adjust the Wasserstein metric to accelerate convergence. Extensive numerical results showed that Ada-GWG outperforms conventional ParVI methods.

## Acknowledgements

This work was supported by National Natural Science Foundation of China (grant no. 12201014 and grant no. 12292983). The research of Cheng Zhang was supported in part by National Engineering Laboratory for Big Data Analysis and Applications, the Key Laboratory of Mathematics and Its Applications (LMAM) and the Key Laboratory of Mathematical Economics and Quantitative Finance (LMEQF) of Peking University. Ziheng Cheng and Shiyue Zhang are partially supported by the elite undergraduate training program of School of Mathematical Sciences in Peking University. The authors are grateful for the computational resources provided by the High-performance Computing Platform of Peking University. The authors appreciate the anonymous NeurIPS reviewers for their constructive feedback.

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

# A  Minimizing Movement Scheme

## A.1  Geometric Interpretation

In fact, under some mild conditions of $g$, the transportation cost $W_{c_h}(\cdot, \cdot)$ can induce a Wasserstein metric and thus $\mathcal{P}_{c_h}(\mathbb{R}^d) := \{\mu \in \mathcal{P}(\mathbb{R}^d) : \mathbb{E}_\mu[g(\frac{2x}{h})] < \infty\}$ is indeed a Wasserstein space.

**Proposition A.1.** *Let $g(\cdot) = g_0(\|\cdot\|)$ where $g_0 : \mathbb{R}^+ \cup \{0\} \to \mathbb{R}^+ \cup \{0\}$ satisfies $g_0(0) = 0$ and $\|\cdot\|$ can be any norm in $\mathbb{R}^d$. Then $g_0^{-1}(W_{c_h}(\cdot, \cdot))$ is a metric on $\mathcal{P}_{c_h}(\mathbb{R}^d)$ if $g_0$ satisfies: (1) $g_0$ is continuous and strictly increasing; (2) $g_0$ is convex; (3) $\log g_0(x)$ is a convex function of $\log x$.*

*Proof.* Suppose $\pi, \mu, \nu \in \mathcal{P}_{c_h}$. It is obvious that $g_0^{-1}(W_{c_h}(\mu, \nu)) = 0$ if and only if $\mu = \nu$. Besides, $g_0^{-1}(W_{c_h}(\cdot, \cdot))$ is symmetric. In the rest part of proof we aim to show that $g_0^{-1}(W_{c_h}(\mu, \pi)) + g_0^{-1}(W_{c_h}(\nu, \pi)) \geq g_0^{-1}(W_{c_h}(\mu, \nu))$. By Gluing lemma Villani et al. (2009), we can construct random variables $X \sim \pi, Y \sim \mu, Z \sim \nu$ such that $(X, Y), (X, Z)$ are the optimal coupling of $(\pi, \mu)$ and $(\pi, \nu)$ for transportation cost $W_{c_h}$, respectively. Then we have

$$
\begin{aligned}
g_0^{-1}(W_{c_h}(\mu, \nu)) &\leq g_0^{-1}\left(\mathbb{E}g_0\left(\frac{\|Y - Z\|}{h}\right)\right) \\
&\leq g_0^{-1}\left(\mathbb{E}g_0\left(\frac{\|X - Y\| + \|X - Z\|}{h}\right)\right) \\
&\leq g_0^{-1}\left(\mathbb{E}g_0\left(\frac{\|X - Y\|}{h}\right)\right) + g_0^{-1}\left(\mathbb{E}g_0\left(\frac{\|X - Z\|}{h}\right)\right) \\
&= g_0^{-1}(W_{c_h}(\mu, \pi)) + g_0^{-1}(W_{c_h}(\pi, \nu)).
\end{aligned}
$$

Here the last inequality is due to generalized Minkowski's inequality Mulholland (1949). $\qquad\square$

The conditions in Proposition A.1 are mild and the most common choices of Young function $g$ satisfy them, and hence can induce a Wasserstein space the generalized Wasserstein gradient flow. Some typical examples of $g_0$ include $|x|^p, \exp(ax^2) - 1, x\exp(ax^b)$, while the norm $\|\cdot\|$ in $\mathbb{R}^d$ can be $\|\cdot\|_p, \|\cdot\|_H$ and so on.

## A.2  Derivation of Generalized Wasserstein Gradient Flow

**Theorem A.2** (Restatement of Theorem 1)**.** *Given a continuously differentiable Young function $g$ and step size $h > 0$, define cost function $c_h(x, y) = g(\frac{x - y}{h})h$. Suppose that $\pi, \mu_{kh} \in \mathcal{P}_{c_h}(\mathbb{R}^d) := \{\mu \in \mathcal{P}(\mathbb{R}^d) : \mathbb{E}_\mu[g(\frac{2x}{h})] < \infty\}$. If $g$ satisfies assumptions in Proposition A.1, $\mathcal{P}_{c_h}(\mathbb{R}^d)$ is a Wasserstein space equipped with Wasserstein metric. Consider MMS under transportation cost $W_{c_h}$:*

$$
\mu_{(k+1)h} := \underset{\mu \in \mathcal{P}_{c_h}(\mathbb{R}^d)}{\arg\min} \ D_{\mathrm{KL}}(\mu\|\pi) + W_{c_h}(\mu, \mu_{kh}). \tag{24}
$$

*Denote the optimal transportation map under $W_{c_h}$ from $\mu_{(k+1)h}$ to $\mu_{kh}$ by $T_k(\cdot)$. Then we have*

$$
\frac{T_k(x) - x}{h} = -\nabla g^*\left(\nabla \log \pi(x) - \nabla \log \mu_{(k+1)h}(x)\right). \tag{25}
$$

*Proof.* By Kantorovich duality (Villani, 2021), the optimal transportation cost (3) has an equivalent definition:

$$
W_c(\mu, \nu) = \sup_\varphi \int \varphi d\mu + \int \varphi^c d\nu, \quad \text{where } \varphi^c(y) := \inf_{x \in \mathbb{R}^d} c(x, y) - \varphi(x). \tag{26}
$$

Take the functional derivative of the optimization problem (24) and define the optimal $\varphi$ in (26) as $\psi$. The following holds:

$$
\frac{\delta}{\delta\mu} D_{\mathrm{KL}}(\mu_{(k+1)h}\|\pi) + \psi_{kh} = \text{const.} \tag{27}
$$

Here the reason for have a constant instead of zero is that we constrain $\mu_{(k+1)h}$ in the space of smooth probability density. Note that $c_h(x,y) \leq \frac{1}{2}\left(g(\frac{2x}{h}) + g(\frac{2y}{h})\right)$ and $\mu_{kh}, \mu_{(k+1)h} \in \mathcal{P}_{c_h}(\mathbb{R}^d)$, then by the fundamental theorem of optimal transportation (Villani, 2021),

$$\psi_{kh}(x) + \psi_{kh}^{c_h}(y) = c_h(x,y), \text{ for } y = T_k(x), \tag{28}$$

which implies $\nabla\psi_{kh}(x) = \nabla_x c_h(x,y) = \nabla g(\frac{x-y}{h})$, i.e.,

$$x - y = \nabla g^*(\nabla\psi_{kh}(x))h. \tag{29}$$

Combine this equation with (27) and thus the optimal map is given by

$$\frac{T_k(x) - x}{h} = -\nabla g^*(-\nabla\frac{\delta}{\delta\mu}D_{\mathrm{KL}}(\mu_{(k+1)h}\|\pi))$$

$$= -\nabla g^*(\nabla\log\frac{\pi}{\mu_{(k+1)h}}).$$

$\square$

# B    Details of Motivating Example

We use Example 1 to illustrate the benefits of choosing general Young function $g^*$, which is also discussed in Balasubramanian et al. (2022); Wibisono & Yang (2022).

We follow the procedures of Wibisono & Yang (2022). For convenience, let $\pi_0 = \mathcal{N}(-m, 1), \pi_1 = \mathcal{N}(m, 1)$ and rewrite $\pi = \frac{1}{2}\pi_0 + \frac{1}{2}\pi_1, \mu = \frac{3}{4}\pi_0 + \frac{1}{4}\pi_1$. The lower bound of KL divergence follows from Devroye et al. (2018) and *Pinsker inequality*. In addition, Balasubramanian et al. (2022) shows

$$\nabla\log\pi - \nabla\log\mu = -m\frac{\pi_0\pi_1}{2\pi\mu}. \tag{30}$$

Also note that $\frac{\pi_0}{\pi_1} = \exp(-2mx)$. Therefore for any $q \geq 1$, the following bound holds:

$$\mathbb{E}_\mu\left\|\nabla\log\frac{\pi}{\mu}\right\|_q^q = \frac{m^q}{2^q}\int\frac{\pi_0^q\pi_1^q}{\mu^{q-1}\pi^q}dx$$

$$= 4^{q-1}m^q\int\frac{\pi_0^q\pi_1^q}{(3\pi_0 + \pi_1)^{q-1}(\pi_0 + \pi_1)^q}dx$$

$$\geq 4^{q-1}m^q\left(\int_{x\geq\frac{1}{m}}\frac{\pi_0^q}{(1+e^{-2})^q\pi_1^{q-1}(1+3e^{-2})^{q-1}}dx + \int_{x\leq-\frac{1}{m}}\frac{\pi_1^q}{(1+e^{-2})^q\pi_0^{q-1}(3+e^{-2})^{q-1}}dx\right). \tag{31}$$

$$\int_{x\geq\frac{1}{m}}\frac{\pi_0^q}{\pi_1^{q-1}}dx = \int_{x\geq\frac{1}{m}}\frac{1}{\sqrt{2\pi}}\exp(-\frac{1}{2}(x+m)^2 - 2m(q-1)x)dx$$

$$= \mathbf{Prob}_{\mathcal{N}(0,1)}\{Z \geq (2q-1)m + \frac{1}{m}\}\exp(2q(q-1)m^2)$$

$$\geq \frac{3}{4}\frac{1}{(2q-1)m + \frac{1}{m}}\frac{1}{\sqrt{2\pi}}\exp\left(-\frac{1}{2}((2q-1)m + \frac{1}{m})^2 + 2q(q-1)m^2\right) \tag{32}$$

$$\geq \frac{3}{4}\frac{1}{2qm}\frac{1}{\sqrt{2\pi}}\exp(-\frac{1}{2}m^2 - 2q + \frac{1}{2}).$$

Here the first inequality is by $\mathbf{Prob}_{\mathcal{N}(0,1)}\{Z \geq t\} \geq (\frac{1}{t} - \frac{1}{t^3})\frac{1}{\sqrt{2\pi}}\exp(-\frac{t^2}{2})$ for any $t > 0$ and $(2q-1)m + \frac{1}{m} \geq 2$. Similarly, we can prove that

$$\int_{x\leq-\frac{1}{m}}\frac{\pi_1^q}{\pi_0^{q-1}}dx \geq \frac{3}{4}\frac{1}{2qm}\frac{1}{\sqrt{2\pi}}\exp(-\frac{1}{2}m^2 - 2q + \frac{1}{2}). \tag{33}$$

Plug (32) and (33) in (31),

$$\mathbb{E}_\mu \left\| \nabla \log \frac{\pi}{\mu} \right\|_q^q \geq \frac{1}{q} m^{q-1} \exp(-\frac{m^2}{2}) \cdot \frac{3}{4^3} \sqrt{\frac{2e}{\pi}} 4^q (e^2+1)^{-q} \left[ (\frac{1}{1+3e^{-2}})^{q-1} + (\frac{1}{3+e^{-2}})^{q-1} \right]$$

$$\geq \frac{1}{q} m^{q-1} \exp(-\frac{m^2}{2}) \cdot \frac{3}{4^3} \sqrt{\frac{2e}{\pi}} (1+3e^{-2}) \left[ \frac{4}{(1+e^2)(1+3e^{-2})} \right]^q$$

$$\geq \frac{0.08}{qm} (\frac{m}{3})^q \exp(-\frac{m^2}{2}).$$

As for the upper bound,

$$\mathbb{E}_\mu \left\| \nabla \log \frac{\pi}{\mu} \right\|_q^q = \frac{m^q}{2^q} \int \frac{\pi_0^q \pi_1^q}{\mu^{q-1} \pi^q} dx$$

$$= 4^{q-1} m^q \int \frac{\pi_0^q \pi_1^q}{(3\pi_0 + \pi_1)^{q-1} (\pi_0 + \pi_1)^q} dx \tag{34}$$

$$\leq 4^{q-1} m^q \left( \int_{x \geq 0} \frac{\pi_0^q}{\pi_1^{q-1}} dx + \int_{x \leq 0} \frac{\pi_1^q}{\pi_0^{q-1}} dx \right).$$

$$\int_{x \geq 0} \frac{\pi_0^q}{\pi_1^{q-1}} dx = \int_{x \geq 0} \frac{1}{\sqrt{2\pi}} \exp(-\frac{1}{2}(x+m)^2 - 2m(q-1)x) dx$$

$$= \mathbf{Prob}_{\mathcal{N}(0,1)} \{ Z \geq (2q-1)m \} \exp(2q(q-1)m^2)$$

$$\leq \frac{1}{(2q-1)m} \frac{1}{\sqrt{2\pi}} \exp\left( -\frac{1}{2}((2q-1)m)^2 + 2q(q-1)m^2 \right) \tag{35}$$

$$\leq \frac{1}{qm} \frac{1}{\sqrt{2\pi}} \exp(-\frac{1}{2}m^2).$$

Here the first inequality is by $\mathbf{Prob}_{\mathcal{N}(0,1)} \{ Z \geq t \} \leq \frac{1}{t} \frac{1}{\sqrt{2\pi}} \exp(-\frac{t^2}{2})$ for any $t > 0$. Similarly, we can prove that

$$\int_{x \leq 0} \frac{\pi_1^q}{\pi_0^{q-1}} dx \leq \frac{1}{qm} \frac{1}{\sqrt{2\pi}} \exp(-\frac{1}{2}m^2). \tag{36}$$

Therefore

$$\mathbb{E}_\mu \left\| \nabla \log \frac{\pi}{\mu} \right\|_q^q \leq 4^{q-1} m^q \cdot \frac{2}{qm} \frac{1}{\sqrt{2\pi}} \exp(-\frac{1}{2}m^2)$$

$$\leq \frac{0.2}{qm} (4m)^q \exp(-\frac{m^2}{2}).$$

## C  Asymptotic Normality of Estimator

In practice, there are finite particles and we can only get a Monte Carlo estimation of (13). But our theoretical analysis is based on the population loss. With this concern, we show that the maximum point of estimation (14) have good statistical properties. To be specific, the estimator converges to true maximum point with asymptotic normality under mild conditions. Similar properties are also studied in Barp et al. (2019); Song et al. (2020); Koehler et al. (2023).

Define objective function $\ell(w, x) := \nabla \log \pi(x)^T f_w(x) + \nabla_x \cdot f_w(x) - g(f_w(x))$.

**Assumption 4.** $\mathcal{W}$ is compact and $\mathcal{L}(\cdot)$ defined in (13) has a unique maximum point $w^* \in int(\mathcal{W})$.

**Assumption 5.** $f_w(x), \nabla_x f_w(x)$ are continuous with $w$. $\max\{\|f_w(x)\|, |\nabla_x \cdot f_w(x)|, g(f_w(x))\} \leq M_0(x)$ for some $M_0 \in \mathcal{L}^1(\mu)$.

**Assumption 6.** There exists a neighborhood $\mathcal{N}$ of $w^*$ such that $\ell(\cdot, x)$ is twice differentiable in $\mathcal{N}$ and $\|\nabla^2 \ell(w, x)\| \leq M_1(x)$ for all $w \in \mathcal{N}$. Additionally, assume $M_1 \in \mathcal{L}^1(\mu)$ and $H := \nabla^2 \mathcal{L}(w^*)$ is non-singular.

**Theorem C.1.** *Given $x_1, \cdots, x_n \overset{i.i.d.}{\sim} \mu$, let $\widehat{w}_n := \arg\min\limits_{w} \widehat{\mathcal{L}}_n(w) := \frac{1}{n}\sum\limits_{i=1}^{n} \ell(w, x_i)$. Under*
*Assumption 4-6, we have*

$$\sqrt{n}(\widehat{w}_n - w^*) \overset{p}{\to} \mathcal{N}\left(0, H^{-1}\Sigma H^{-1}\right), \tag{37}$$

*where $\Sigma = \mathbb{E}_\mu \nabla_w \ell(w^*, \cdot) \otimes \nabla_w \ell(w^*, \cdot)$.*

*Proof.* Note that under Assumption 5, $|\ell(w, x)| \leq M(x)$ for some $M \in \mathcal{L}^1(\mu)$. By Newey & McFadden (1986, Lemma 2.4, Theorem 2.1), $\widehat{w}_n$ is weakly consistent for $w^*$. Additionally, since $w^*$ is the maximum point of $\mathcal{L}$ and by Central Limit Theorem,

$$\sqrt{n}\nabla\widehat{\mathcal{L}}_n(w^*) = \frac{1}{\sqrt{n}}\sum_{i=1}^{n} \nabla_w \ell(w^*, x_i) \overset{p}{\to} \mathcal{N}(0, \Sigma).$$

By Assumption 6 and Newey & McFadden (1986, Lemma 2.4), the second order derivative converges uniformly, *i.e.*

$$\sup_{w \in \mathcal{N}} \|\nabla^2 \widehat{\mathcal{L}}_n(w) - \nabla^2 \mathcal{L}(w)\| \overset{p}{\to} 0.$$

Finally, the result follows Newey & McFadden (1986, Theorem 3.1).

$\square$

# D   Proof in Section 4

We first justify our Assumption 1 and 2, then present some crucial lemmas and finish the proof of convergence results. Our proof procedure uses interpolation process of discrete dynamics, following Vempala & Wibisono (2019); Balasubramanian et al. (2022). Informally, the difference between discrete dynamics and continuous dynamics consists of two parts: discretization error and estimation error (by neural nets). We bound the discretization error in Lemma D.3, D.4 and the estimation error in Lemma D.5, D.6.

## D.1   Justification for Assumption 1 and 2

*Proof of Proposition 2.* It suffices to show that there exists $\varepsilon_2 < \infty$ such that for any $a, b \in \mathbb{R}$, the following inequality holds:

$$|b - \text{sgn}(a)|a|^{q-1}|^p \leq \varepsilon_1 \frac{|a|^q}{q} + \varepsilon_2 \left(\frac{|a|^q}{q} - ab + \frac{|b|^p}{p}\right). \tag{38}$$

If $a = 0$, then $\varepsilon_2 \geq p$ is sufficient. Without loss of generality, suppose $a = 1$ (by replacing $b$ with $\text{sgn}(a)b/|a|^{q-1}$). We only need to show that

$$|b - 1|^p \leq \varepsilon_1 + \varepsilon_2 \left(\frac{1}{q} - b + \frac{|b|^p}{p}\right). \tag{39}$$

(1) **Case $p \geq 2$.**

Let $\varepsilon_1 = 0$. Since $\lim\limits_{b \to 1} \dfrac{|b-1|^p}{\frac{1}{q} - b + \frac{|b|^p}{p}} = \lim\limits_{b \to 1} \dfrac{2|b-1|^{p-2}}{p-1} \leq 2$, so there exists $\delta > 0$ such that

when $b \in [1 - \delta, 1 + \delta]$, (39) holds if $\varepsilon_2 \geq p + 1 > 2$. Also note that $f(b) = \dfrac{|b-1|^p}{\frac{1}{q} - b + \frac{|b|^p}{p}}$ is a

continuous function on $\mathbb{R}\setminus(1 - \delta, 1 + \delta)$ and $\lim\limits_{b \to \infty} f(b) = p < +\infty$. Therefore, $f(b)$ is bounded on $\mathbb{R}\setminus[1 - \delta, 1 + \delta]$ and thus (39) holds for finite $\varepsilon_2$. It's obvious that $\varepsilon_2$ only depends on $p$ in this case.

(2) **Case $p < 2$.**

Similarly, let $\delta = \varepsilon_1^{1/p}$. When $b \in [1 - \delta, 1 + \delta]$, (39) will trivially hold for any $\varepsilon_2 > 0$. Also, $f(b)$ is bounded on $\mathbb{R}\setminus[1 - \delta, 1 + \delta]$ and thus there exists finite $\varepsilon_2$ determined by $p, \delta$ such that (39) holds.

$\square$

## D.2 Main Lemmas

**Lemma D.1.** *For any $t \in (kh, (k+1)h)$, $\partial_t D_{\mathrm{KL}}(\mu_t \| \pi) = -\mathbb{E}_{\mu_t} \left\langle \nabla \log \frac{\pi}{\mu_t}, \mathbb{E}[v_k(X_{kh})|X_t = \cdot] \right\rangle$*

*Proof.* Let $\mu_{t|\mathcal{F}_{kh}}$ denote the law of $X_t$ conditioned on the filtration $\mathcal{F}_{kh}$ at time $kh$. Then by Fokker-Planck equation, we have

$$\partial_t \mu_{t|\mathcal{F}_{kh}} = -\mathrm{div}\left(\mu_{t|\mathcal{F}_{kh}} v_k(X_{kh})\right).$$

Then we take expectation of the above equation; by Bayesian formula (Vempala & Wibisono, 2019),

$$\partial_t \mu_t = -\mathrm{div}\,\mathbb{E}[\mu_{t|\mathcal{F}_{kh}} v_k(X_{kh})]$$
$$= -\mathrm{div}\left(\mu_t \mathbb{E}[v_k(X_{kh})|X_t = \cdot]\right)$$

Hence

$$\partial_t D_{\mathrm{KL}}(\mu_t \| \pi) = -\mathbb{E}_{\mu_t} \left\langle \nabla \log \frac{\pi}{\mu_t}, \mathbb{E}[v_k(X_{kh})|X_t = \cdot] \right\rangle.$$

$\square$

**Lemma D.2.** *Suppose that $h < \frac{1}{G_2}$. Under Assumption 3, for any $t \in [kh, (k+1)h]$, $q > 1, \frac{1}{p} + \frac{1}{q} = 1$,*

$$\mathbb{E}_{\mu_t} \left\| \nabla \log \frac{\mu_t}{\mu_{kh}} \right\|_q^q \leq \left( M_p (1 - hG_2)^{-1} dh \right)^q$$

*Proof.* Note that $\log \frac{\mu_t}{\mu_{kh}} = -\log\det\left(I_d + (t-kh)\nabla v_k\right)$ since $x \mapsto x + (t-kh)v_k(x)$ is an orientation-preserving diffeomorphism under $h < \frac{1}{G_2}$. Then the following holds:

$$\left\| \nabla \log \frac{\mu_t}{\mu_{kh}}(x) \right\|_q = \sup_{\|z\|_p = 1} \left\langle \nabla \log \frac{\mu_t}{\mu_{kh}}(x), z \right\rangle$$

$$= \sup_{\|z\|_p = 1} \lim_{\delta \to 0} \left| \log\det\left(I_d + (t-kh)\nabla v_k(x + \delta z)\right) - \log\det\left(I_d + (t-kh)\nabla v_k(x)\right) \right|$$

$$\leq \sup_{\|z\|_p = 1} \lim_{\delta \to 0} (t-kh)\|\nabla v_k(x + \delta z) - \nabla v_k(x)\|_2 (1 - hG_2)^{-1} d$$

$$\leq M_p (1 - hG_2)^{-1} dh.$$

Here the first equation is due to Young's inequality and the second equation follows the definition of gradient. The inequality in the third line is due to Lemma D.13 and the last one is by Assumption 3. With this uniform bound we finish the proof. $\square$

**Lemma D.3.** *Under Assumption 1, 3, and the same conditions in Lemma D.2,*

$$\mathbb{E}_{\mu_t} \left\| \nabla g^*(\nabla \log \frac{\pi}{\mu_t}) - \nabla g^*(\nabla \log \frac{\pi}{\mu_{kh}}) \right\|_p^p \leq c_1 \mathbb{E}_{\mu_t} \left\| \nabla \log \frac{\pi}{\mu_t} \right\|_q^q + c_2 \left( M_p (1 - hG_2)^{-1} dh \right)^q$$

*where $c_1, c_2$ are defined as:*

$$(c_1, c_2) = \begin{cases} (0, \ 2^{p-q}) & \text{if } q \leq 2, \\ \left( 3^{-p}, \ \min\left\{ 3^{q-p}(\frac{p}{q})^{\frac{q-p}{q-1}}(1 - \frac{p}{q})^{\frac{q-p}{p}}, (q-1)^p \left( \frac{(\frac{4}{3})^{\frac{1}{q-1}}}{(\frac{4}{3})^{\frac{1}{q-1}} - 1} \right)^{q-p} \right\} \right) & \text{otherwise.} \end{cases}$$

$$(40)$$

*Proof.* Since $g^*(x) = \dfrac{1}{q}\|x\|_q^q$, we have $\nabla g^*(x) = \text{sgn}(x) \odot |x|^{q-1}$. Here $\odot$ means entry-wise product. Apply Lemma D.14 entry-wise and thus

$$\mathbb{E}_{\mu_t}\left\|\nabla g^*(\nabla \log \frac{\pi}{\mu_t}) - \nabla g^*(\nabla \log \frac{\pi}{\mu_{kh}})\right\|_p^p \leq c_1\mathbb{E}_{\mu_t}\left\|\nabla \log \frac{\pi}{\mu_t}\right\|_q^q + c_2\mathbb{E}_{\mu_t}\left\|\nabla \log \frac{\mu_t}{\mu_{kh}}\right\|_q^q$$

$$\leq c_1\mathbb{E}_{\mu_t}\left\|\nabla \log \frac{\pi}{\mu_t}\right\|_q^q + c_2\left(M_p(1 - hG_2)^{-1}dh\right)^q.$$

The second inequality is due to Lemma D.2. $\qquad\square$

**Lemma D.4.** *Under Assumption 2, 3 and the same conditions in Lemma D.2,*

$$\mathbb{E}_{\mu_t}\left\|\nabla g^*(\nabla \log \frac{\pi}{\mu_t}) - \nabla g^*(\nabla \log \frac{\pi}{\mu_{kh}})\right\|_2^2 \leq \beta^2\left(M_2(1 - hG_2)^{-1}dh\right)^2$$

*Proof.* Note that $g^*$ is $\beta$-smooth and by Lemma D.2,

$$\mathbb{E}_{\mu_t}\left\|\nabla g^*(\nabla \log \frac{\pi}{\mu_t}) - \nabla g^*(\nabla \log \frac{\pi}{\mu_{kh}})\right\|_2^2 \leq \beta^2\mathbb{E}_{\mu_t}\left\|\nabla \log \frac{\mu_t}{\mu_{kh}}\right\|_2^2$$

$$\leq \beta^2\left(M_2(1 - hG_2)^{-1}dh\right)^2.$$

$\qquad\square$

**Lemma D.5.** *Suppose that $h < \min\left\{\dfrac{1}{4G_p}, \dfrac{1}{G_2}\right\}$. Under Assumption 1, 3, for any $t \in [kh, (k+1)h]$,*

$$\mathbb{E}_{\mu_t}\left\|\nabla g^*(\nabla \log \frac{\pi}{\mu_{kh}}) - \mathbb{E}[v_k(X_{kh})|X_t = \cdot]\right\|_p^p$$

$$\leq \frac{2^{p-1}(1 - hG_2)^{-d}}{1 - (4G_ph)^p}\varepsilon_k + \frac{2^{p-1}(4G_ph)^p}{1 - (4G_ph)^p}\left((1 + c_1)\mathbb{E}_{\mu_t}\left\|\nabla \log \frac{\pi}{\mu_t}\right\|_q^q + c_2\left(M_p(1 - hG_2)^{-1}dh\right)^q\right),$$

*where $c_1, c_2$ are defined in (40).*

*Proof.* By Jensen's inequality,

$$\mathbb{E}_{\mu_t}\left\|\nabla g^*(\nabla \log \frac{\pi}{\mu_{kh}}) - \mathbb{E}[v_k(X_{kh})|X_t = \cdot]\right\|_p^p$$

$$\leq 2^{p-1}\left\{\mathbb{E}_{\mu_t}\left\|\nabla g^*(\nabla \log \frac{\pi}{\mu_{kh}}) - v_k\right\|_p^p + \mathbb{E}_{\mu_t}\left\|v_k - \mathbb{E}[v_k(X_{kh})|X_t = \cdot]\right\|_p^p\right\} \quad (41)$$

$$\leq 2^{p-1}\left\{\mathbb{E}_{\mu_t}\left\|\nabla g^*(\nabla \log \frac{\pi}{\mu_{kh}}) - v_k\right\|_p^p + \mathbb{E}\left\|v_k(X_t) - v_k(X_{kh})\right\|_p^p\right\}.$$

Under Assumption 3 and Jensen's inequality,

$$\mathbb{E}\left\|v_k(X_t) - v_k(X_{kh})\right\|_p^p$$

$$\leq (G_p)^p\,\mathbb{E}\|X_t - X_{kh}\|_p^p$$

$$\leq (G_ph)^p\,\mathbb{E}\left\|v_k(X_{kh})\right\|_p^p$$

$$\leq (G_ph)^p4^{p-1}\left\{\mathbb{E}\left\|v_k(X_{kh}) - v_k(X_t)\right\|_p^p + \mathbb{E}\left\|v_k(X_t) - \nabla g^*\left(\nabla \log \frac{\pi}{\mu_{kh}}(X_t)\right)\right\|_p^p\right.$$

$$\left. + \mathbb{E}\left\|\nabla g^*\left(\nabla \log \frac{\pi}{\mu_{kh}}(X_t)\right) - \nabla g^*\left(\nabla \log \frac{\pi}{\mu_t}(X_t)\right)\right\|_p^p + \mathbb{E}\left\|\nabla g^*\left(\nabla \log \frac{\pi}{\mu_t}(X_t)\right)\right\|_p^p\right\}.$$

Rearrange the above inequality and thus

$$\mathbb{E}\left|\left|v_k(X_t) - v_k(X_{kh})\right|\right|_p^p$$

$$\leq (1 - (4G_p h)^p)^{-1} (4G_p h)^p \left\{ \mathbb{E}_{\mu_t}\left|\left|v_k - \nabla g^*(\nabla \log \frac{\pi}{\mu_{kh}})\right|\right|_p^p \right. \tag{42}$$

$$\left. + \mathbb{E}_{\mu_t}\left|\left|\nabla g^* \nabla \log \frac{\pi}{\mu_{kh}}) - \nabla g^*(\nabla \log \frac{\pi}{\mu_t})\right|\right|_p^p + \mathbb{E}_{\mu_t}\left|\left|\nabla g^*(\nabla \log \frac{\pi}{\mu_t})\right|\right|_p^p \right\}.$$

Again note that $\dfrac{\mu_t}{\mu_{kh}} = \det \left(I_d + (t - kh)\nabla v_k\right)^{-1}$, so

$$\sup_x \frac{\mu_t}{\mu_{kh}}(x) \leq \left(\|(I_d + (t - kh)\nabla v_k(x))^{-1}\|_2\right)^d \leq (1 - hG_2)^{-d}.$$

Then by Assumption 1,

$$\mathbb{E}_{\mu_t}\left|\left|v_k - \nabla g^*(\nabla \log \frac{\pi}{\mu_{kh}})\right|\right|_p^p \leq (1 - hG_2)^{-d}\mathbb{E}_{\mu_{kh}}\left|\left|v_k - \nabla g^*(\nabla \log \frac{\pi}{\mu_{kh}})\right|\right|_p^p \leq (1 - hG_2)^{-d}\varepsilon_k.$$

$$\tag{43}$$

Combining Lemma D.3 with (42),(43) and plugging them into (41), we finish the proof. $\qquad\square$

**Lemma D.6.** *Suppose that* $h < \dfrac{1}{4G_2}$*. Under Assumption 2, 3, for any* $t \in [kh, (k+1)h]$*,*

$$\mathbb{E}_{\mu_t}\left|\left|\nabla g^*(\nabla \log \frac{\pi}{\mu_{kh}}) - \mathbb{E}[v_k(X_{kh})|X_t = \cdot]\right|\right|_2^2$$

$$\leq \frac{2(1 - hG_2)^{-d}}{1 - (4G_2 h)^2}\varepsilon_k + \frac{2(4G_2 h)^2\beta^2}{1 - (4G_2 h)^2}\left(\mathbb{E}_{\mu_t}\left|\left|\nabla \log \frac{\pi}{\mu_t}\right|\right|_2^2 + \left(M_2(1 - hG_2)^{-1}dh\right)^2\right).$$

*Proof.* The procedure is exactly the same with Lemma D.5. The only difference appears when applying Lemma D.4 instead of Lemma D.3 in the last step. $\qquad\square$

**Lemma D.7.** *Suppose that* $h \leq \min\left\{\dfrac{1}{36G_p}, \dfrac{1 - 2^{-\frac{1}{q}}}{G_2}\right\}$*. Under Assumption 1, 3, for any* $t \in (kh, (k+1)h)$*,*

$$\partial_t D_{\mathrm{KL}}(\mu_t \| \pi) \leq -\frac{1}{12}\mathbb{E}_{\mu_t}\left|\left|\nabla \log \frac{\pi}{\mu_t}\right|\right|_q^q + A_1(M_p dh)^q + A_2(1 - hG_2)^{-d}\varepsilon_k,$$

*where* $A_1, A_2$ *are constants only depending on* $q$*:*

$$(A_1, A_2) = \begin{cases} (\dfrac{2^{p+2}}{p}, 2^p) & \textit{if } q \leq 2, \\ (\dfrac{7c_2}{p}, \dfrac{3}{p}) & \textit{otherwise.} \end{cases} \tag{44}$$

*Here* $c_2$ *is defined in (40).*

*Proof.* By Lemma D.1 and Young's inequality, for any $\lambda_1, \lambda_2 > 0$,

$$\partial_t D_{\mathrm{KL}}(\mu_t \| \pi) = -\mathbb{E}_{\mu_t} \left\langle \nabla \log \frac{\pi}{\mu_t}, \nabla g^*(\nabla \log \frac{\pi}{\mu_t}) \right\rangle$$

$$+ \mathbb{E}_{\mu_t} \left\langle \nabla \log \frac{\pi}{\mu_t}, \nabla g^*(\nabla \log \frac{\pi}{\mu_t}) - \nabla g^*(\nabla \log \frac{\pi}{\mu_{kh}}) \right\rangle$$

$$+ \mathbb{E}_{\mu_t} \left\langle \nabla \log \frac{\pi}{\mu_t}, \nabla g^*(\nabla \log \frac{\pi}{\mu_{kh}}) - \mathbb{E}[v_k(X_{kh})|X_t = \cdot] \right\rangle$$

$$\leq -(1 - \frac{1}{q}\lambda_1^q - \frac{1}{q}\lambda_2^q)\mathbb{E}_{\mu_t} \left\| \nabla \log \frac{\pi}{\mu_t} \right\|_q^q$$

$$+ \frac{1}{p}\lambda_1^{-p}\mathbb{E}_{\mu_t} \left\| \nabla g^*(\nabla \log \frac{\pi}{\mu_t}) - \nabla g^*(\nabla \log \frac{\pi}{\mu_{kh}}) \right\|_p^p$$

$$+ \frac{1}{p}\lambda_2^{-p}\mathbb{E}_{\mu_t} \left\| \nabla g^*(\nabla \log \frac{\pi}{\mu_{kh}}) - \mathbb{E}[v_k(X_{kh})|X_t = \cdot] \right\|_p^p.$$

Then we apply Lemma D.3 and Lemma D.5,

$$\partial_t D_{\mathrm{KL}}(\mu_t \| \pi) \leq -\left(1 - \frac{1}{q}\lambda_1^q - \frac{1}{p}c_1\lambda_1^{-p} - \frac{1}{q}\lambda_2^q - \frac{1}{p}\lambda_2^{-p}\frac{2^{p-1}(4G_p h)^p}{1 - (4G_p h)^p}(1 + c_1)\right)\mathbb{E}_{\mu_t} \left\| \nabla \log \frac{\pi}{\mu_t} \right\|_q^q$$

$$+ \frac{c_2}{p}\left(\lambda_1^{-p} + \lambda_2^{-p}\frac{2^{p-1}(4G_p h)^p}{1 - (4G_p h)^p}\right)\left(M_p(1 - hG_2)^{-1}dh\right)^q + \frac{\lambda_2^{-p}}{p}\frac{2^{p-1}(1 - hG_2)^{-d}}{1 - (4G_p h)^p}\varepsilon_k.$$

If $q > 2$ so that $c_1 = 3^{-p} < 1$, take $\lambda_1 = c_1^{\frac{1}{p+q}}, \lambda_2 = 1$. Note that for $h \leq \frac{1}{36G_p}, \frac{2^{p-1}(4G_p h)^p}{1 - (4G_p h)^p}(1 + c_1) \leq \frac{4G_p h}{1 - 4G_p h} \cdot \frac{4}{3} \leq \frac{1}{6}$. And thus,

$$\partial_t D_{\mathrm{KL}}(\mu_t \| \pi) \leq -\left(\frac{2}{3} - \frac{1}{q} - \frac{1}{p} \cdot \frac{1}{6}\right)\mathbb{E}_{\mu_t} \left\| \nabla \log \frac{\pi}{\mu_t} \right\|_q^q + \frac{7c_2}{p}(M_p dh)^q + \frac{3}{p}(1 - hG_2)^{-d}\varepsilon_k$$

$$\leq -(\frac{5}{6p} - \frac{1}{3})\mathbb{E}_{\mu_t} \left\| \nabla \log \frac{\pi}{\mu_t} \right\|_q^q + \frac{7c_2}{p}(M_p dh)^q + \frac{3}{p}(1 - hG_2)^{-d}\varepsilon_k$$

$$\leq -\frac{1}{12}\mathbb{E}_{\mu_t} \left\| \nabla \log \frac{\pi}{\mu_t} \right\|_q^q + \frac{7c_2}{p}(M_p dh)^q + \frac{3}{p}(1 - hG_2)^{-d}\varepsilon_k.$$

If $q \leq 2$ so that $c_1 = 0$, take $\lambda_1 = \lambda_2 = (\frac{q}{3})^{\frac{1}{q}}$. Note that for $h \leq \frac{1}{36G_p}, \frac{2^{p-1}(4G_p h)^p}{1 - (4G_p h)^p}(1 + c_1) \leq \frac{(8G_p h)^2}{2(1 - 4G_p h)} \leq \frac{12}{5}G_p h \leq \frac{1}{15}$. And thus,

$$\partial_t D_{\mathrm{KL}}(\mu_t \| \pi) \leq -\left(\frac{1}{3} - \frac{4}{3p}(\frac{3}{q})^{\frac{q}{p}}G_p h\right)\mathbb{E}_{\mu_t} \left\| \nabla \log \frac{\pi}{\mu_t} \right\|_q^q + \frac{c_2 q}{p}(\frac{3}{q})^q(M_p dh)^q + \frac{2^p}{p}\frac{3}{p}(\frac{3}{p})^{\frac{q}{p}}(1 - hG_2)^{-d}\varepsilon_k$$

$$\leq -\left(\frac{1}{3} - \frac{4}{p}G_p h\right)\mathbb{E}_{\mu_t} \left\| \nabla \log \frac{\pi}{\mu_t} \right\|_q^q + \frac{2^{p+2}}{p}(M_p dh)^q + 2^p(1 - hG_2)^{-d}\varepsilon_k$$

$$\leq -\frac{1}{6}\mathbb{E}_{\mu_t} \left\| \nabla \log \frac{\pi}{\mu_t} \right\|_q^q + \frac{2^{p+2}}{p}(M_p dh)^q + 2^p(1 - hG_2)^{-d}\varepsilon_k.$$

Therefore, define $A_1, A_2$ as in (44) and we finish the proof. $\square$

**Lemma D.8.** *Suppose that* $h \leq \dfrac{1}{4G_2\sqrt{12\kappa^2 + 1}}$, *where* $\kappa := \dfrac{\beta}{\alpha} \geq 1$. *Under Assumption 2, 3, for any* $t \in (kh, (k+1)h)$,

$$\partial_t D_{\mathrm{KL}}(\mu_t \| \pi) \leq -\frac{\alpha}{6}\mathbb{E}_{\mu_t} \left\| \nabla \log \frac{\pi}{\mu_t} \right\|_2^2 + \frac{3}{\alpha}(\beta M_2 dh)^2 + \frac{4}{\alpha}(1 - hG_2)^{-d}\varepsilon_k.$$

*Proof.* Similar to Lemma D.7, under $g^*$ is $\alpha$-strongly convex,

$$\partial_t D_{\mathrm{KL}}(\mu_t \| \pi) = -\mathbb{E}_{\mu_t} \left\langle \nabla \log \frac{\pi}{\mu_t}, \nabla g^*(\nabla \log \frac{\pi}{\mu_t}) \right\rangle$$

$$+ \mathbb{E}_{\mu_t} \left\langle \nabla \log \frac{\pi}{\mu_t}, \nabla g^*(\nabla \log \frac{\pi}{\mu_t}) - \nabla g^*(\nabla \log \frac{\pi}{\mu_{kh}}) \right\rangle$$

$$+ \mathbb{E}_{\mu_t} \left\langle \nabla \log \frac{\pi}{\mu_t}, \nabla g^*(\nabla \log \frac{\pi}{\mu_{kh}}) - \mathbb{E}[v_k(X_{kh})|X_t = \cdot] \right\rangle$$

$$\leq -(\alpha - \frac{1}{2}\lambda_1^2 - \frac{1}{2}\lambda_2^2)\mathbb{E}_{\mu_t} \left\| \nabla \log \frac{\pi}{\mu_t} \right\|_2^2$$

$$+ \frac{1}{2}\lambda_1^{-2}\mathbb{E}_{\mu_t} \left\| \nabla g^*(\nabla \log \frac{\pi}{\mu_t}) - \nabla g^*(\nabla \log \frac{\pi}{\mu_{kh}}) \right\|_2^2$$

$$+ \frac{1}{2}\lambda_2^{-2}\mathbb{E}_{\mu_t} \left\| \nabla g^*(\nabla \log \frac{\pi}{\mu_{kh}}) - \mathbb{E}[v_k(X_{kh})|X_t = \cdot] \right\|_2^2.$$

Then we apply Lemma D.4 and Lemma D.6,

$$\partial_t D_{\mathrm{KL}}(\mu_t \| \pi) \leq -\left( \alpha - \frac{1}{2}\lambda_1^2 - \frac{1}{2}\lambda_2^2 - \lambda_2^{-2} \frac{(4G_2 h)^2 \beta^2}{1 - (4G_2 h)^2} \right) \mathbb{E}_{\mu_t} \left\| \nabla \log \frac{\pi}{\mu_t} \right\|_2^2$$

$$+ \frac{1}{2}\left( \lambda_1^{-2} + \lambda_2^{-2} \frac{2(4G_2 h)^2}{1 - (4G_2 h)^2} \right) \left( \beta M_2 (1 - hG_2)^{-1} dh \right)^2 + \frac{\lambda_2^{-2}}{2} \frac{2(1 - hG_2)^{-d}}{1 - (4G_2 h)^2} \varepsilon_k.$$

Take $\lambda_1 = \lambda_2 = \sqrt{\frac{\alpha}{2}}$. Note that for $4G_2 h \leq \frac{1}{\sqrt{12\kappa^2 + 1}}$, we have $\frac{(4G_2 h)^2}{1 - (4G_2 h)^2} \leq \frac{1}{12\kappa^2}$. And thus,

$$\partial_t D_{\mathrm{KL}}(\mu_t \| \pi) \leq -\frac{\alpha}{6}\mathbb{E}_{\mu_t} \left\| \nabla \log \frac{\pi}{\mu_t} \right\|_2^2 + \frac{3}{\alpha}(\beta M_2 dh)^2 + \frac{4}{\alpha}(1 - hG_2)^{-d}\varepsilon_k.$$

$\square$

### D.3 Proof of Main Results

**Theorem D.9.** *Under Assumption 1, 3, for any step size $h \leq \min\left\{ \frac{1}{36G_p}, \frac{1 - 2^{-\frac{1}{q}}}{G_2} \right\}$, it holds that*

$$\frac{1}{Nh} \int_0^{Nh} \mathbb{E}_{\mu_t} \left\| \nabla \log \frac{\pi}{\mu_t} \right\|_q^q dt \leq 12 \left( \frac{D_{\mathrm{KL}}(\mu_0 \| \pi)}{Nh} + A_1 (M_p dh)^q + A_2 (1 - hG_2)^{-d} \frac{\sum_{k=0}^{N-1} \varepsilon_k}{N} \right),$$

*where $A_1, A_2$ defined in (44) are constants that only depend on $q$.*

*Additionally, if $D_{\mathrm{KL}}(\mu_0 \| \pi) \leq K_0$, then for $N \gtrsim \frac{K_0 (G_p \vee (qG_2))^{q+1}}{qA_1 (M_p d)^q}$, we can choose $h \asymp (\frac{K_0}{qA_1(M_p d)^q N})^{\frac{1}{q+1}} \wedge \frac{1}{dG_2}$. The following bound holds:*

$$\mathbb{E}_{\bar{\mu}_{Nh}} \left\| \nabla \log \frac{\pi}{\bar{\mu}_{Nh}} \right\|_q^q = \tilde{\mathcal{O}} \left( (\frac{M_p K_0 d}{N})^{\frac{q}{q+1}} + \frac{G_2 K_0 d}{N} + \frac{\sum_{k=0}^{N-1} \varepsilon_k}{N} \right).$$

*Here $\tilde{\mathcal{O}}(\cdot)$ hides all the constant factors that only depend on $q$.*

*Proof.* Under Lemma D.7, take integral of both sides from $kh$ to $(k+1)h$ and we obtain

$$D_{\mathrm{KL}}(\mu_{(k+1)h} \| \pi) - D_{\mathrm{KL}}(\mu_{kh} \| \pi) \leq -\frac{1}{12} \int_{kh}^{(k+1)h} \mathbb{E}_{\mu_t} \left\| \nabla \log \frac{\pi}{\mu_t} \right\|_q^q dt + A_1 (M_p dh)^q h + A_2 (1 - hG_2)^{-d} \varepsilon_k h.$$

Rearranging it and summing from 0 to $N-1$,

$$\frac{1}{Nh}\int_0^{Nh}\mathbb{E}_{\mu_t}\left\|\nabla\log\frac{\pi}{\mu_t}\right\|_q^q dt \le 12\left(\frac{D_{\mathrm{KL}}(\mu_0\|\pi)}{Nh} + A_1(M_p dh)^q + A_2(1-hG_2)^{-d}\frac{\sum_{k=0}^{N-1}\varepsilon_k}{N}\right).$$
(45)

Note that for any convex function $g^*$ on $\mathbb{R}^d$, $(a,b)\mapsto g^*(a/b)b$ is also convex on $\mathbb{R}^d\times\mathbb{R}_+$. Therefore, $\mu\mapsto\mathbb{E}_\mu g^*(\nabla\log\frac{\pi}{\mu})$ is convex in the classical sense on the space of probability measures. And thus

$$\mathbb{E}_{\bar{\mu}_{Nh}}\left\|\nabla\log\frac{\pi}{\bar{\mu}_{Nh}}\right\|_q^q \le \frac{1}{Nh}\int_0^{Nh}\mathbb{E}_{\mu_t}\left\|\nabla\log\frac{\pi}{\mu_t}\right\|_q^q dt.$$
(46)

We finish the proof by plugging the step size $h$ in (45) and hiding all the constants that only depend on $q$. $\qquad\square$

**Theorem D.10.** *Under Assumption 2, 3, for any step size* $h \le \dfrac{1}{4G_2\sqrt{12\kappa^2+1}}$, *where* $\kappa := \dfrac{\beta}{\alpha} \ge 1$, *it holds that*

$$\frac{1}{Nh}\int_0^{Nh}\mathbb{E}_{\mu_t}\left\|\nabla\log\frac{\pi}{\mu_t}\right\|_2^2 dt \le \frac{6}{\alpha}\left(\frac{D_{\mathrm{KL}}(\mu_0\|\pi)}{Nh} + \frac{3}{\alpha}\beta^2 M_2^2 d^2 h^2 + \frac{4}{\alpha}(1-hG_2)^{-d}\frac{\sum_{k=0}^{N-1}\varepsilon_k}{N}\right).$$

*For simplicity, assume* $\alpha=1$. *If and* $D_{\mathrm{KL}}(\mu_0\|\pi)\le K_0$, *then we can choose* $h\asymp(\dfrac{K_0}{(\kappa M_2 d)^2 N})^{\frac{1}{3}}\wedge\dfrac{1}{dG_2}\wedge\dfrac{1}{\kappa G_2}$. *The following bound holds:*

$$\mathbb{E}_{\bar{\mu}_{Nh}}\left\|\nabla\log\frac{\pi}{\bar{\mu}_{Nh}}\right\|_2^2 = \mathcal{O}\left((\frac{\kappa M_2 K_0 d}{N})^{\frac{2}{3}} + \frac{G_2 K_0(d+\kappa)}{N} + \frac{\sum_{k=0}^{N-1}\varepsilon_k}{N}\right).$$

*Proof.* Under Lemma D.8, take integral of both sides from $kh$ to $(k+1)h$ and we obtain

$$D_{\mathrm{KL}}(\mu_{(k+1)h}\|\pi)-D_{\mathrm{KL}}(\mu_{kh}\|\pi) \le -\frac{\alpha}{6}\int_{kh}^{(k+1)h}\mathbb{E}_{\mu_t}\left\|\nabla\log\frac{\pi}{\mu_t}\right\|_2^2 dt + \frac{3}{\alpha}\beta^2 M_2^2 d^2 h^3 + \frac{4}{\alpha}(1-hG_2)^{-d}\varepsilon_k h.$$

Rearranging it and summing from 0 to $N-1$,

$$\frac{1}{Nh}\int_0^{Nh}\mathbb{E}_{\mu_t}\left\|\nabla\log\frac{\pi}{\mu_t}\right\|_2^2 dt \le \frac{6}{\alpha}\left(\frac{D_{\mathrm{KL}}(\mu_0\|\pi)}{Nh} + \frac{3}{\alpha}\beta^2 M_2^2 d^2 h^2 + \frac{4}{\alpha}(1-hG_2)^{-d}\frac{\sum_{k=0}^{N-1}\varepsilon_k}{N}\right).$$

The remaining part is similar to the proof of Theorem D.9. $\qquad\square$

## D.4 Discussions

The convergence of score divergence only guarantees that the particle distribution gets the local structure of $\pi$ correct (Balasubramanian et al., 2022). To obtain a stronger convergence guarantee, we still need isoperimetry condition of target distribution. We start with $L_2$-GF.

**Theorem D.11.** *If we additionally assume that* $\pi$ *satisfies log-Sobolev inequality with constant* $\lambda$, *i.e.*

$$Ent_\pi(f^2) \le \frac{2}{\lambda}\mathbb{E}_\pi[\|\nabla f\|_2^2], \text{ for all smooth } f:\mathbb{R}^d\to\mathbb{R}.$$

*then under the same conditions of Theorem D.10 with* $\alpha=\beta=1$, $\varepsilon_k\le\epsilon$, *it holds that*

$$D_{\mathrm{KL}}(\mu_{Nh}\|\pi) \le \exp(-\frac{\lambda Nh}{3})D_{\mathrm{KL}}(\mu_0\|\pi) + 3\left(3(M_2 dh)^2 + 4(1-hG_2)^{-d}\epsilon\right)\lambda^{-1}.$$

*In particular, if* $\epsilon\lesssim(\dfrac{M_2}{G_2})^2$, *we take* $h\asymp\dfrac{\sqrt{\epsilon}}{M_2 d}$ *and then we obtain the guarantee* $D_{\mathrm{KL}}(\mu_{Nh}\|\pi)\lesssim\lambda^{-1}\epsilon$ *after*

$$N = \mathcal{O}\left(\frac{M_2 d}{\lambda\sqrt{\epsilon}}\log\frac{\lambda D_{\mathrm{KL}}(\mu_0\|\pi)}{\epsilon}\right) \qquad iterations.$$

**Remark 1.** *We match the SOTA rate of LMC under log-Sobolev inequality, Hessian smoothness and dissipativity assumption (Mou et al., 2022). The assumption on smoothness of target Hessian is known to accelerate convergence rate (Dalalyan & Karagulyan, 2019). But here we do not assume the smoothness of $\log \pi$ explicitly and thus our method can tackle more complex distributions.*

*Proof.* For $t \in (kh, (k+1)h)$, we apply Lemma D.8 and *log-Sobolev inequality* and thus

$$\partial_t D_{\mathrm{KL}}(\mu_t \| \pi) \leq -\frac{\lambda}{3} D_{\mathrm{KL}}(\mu_t \| \pi) + 3(M_2 dh)^2 + 4(1 - hG_2)^{-d} \epsilon. \tag{47}$$

By Gronwall's inequality,

$$D_{\mathrm{KL}}(\mu_{(k+1)h} \| \pi) \leq e^{-\lambda h/3} D_{\mathrm{KL}}(\mu_{kh} \| \pi) + 3\lambda^{-1} \left( 3(M_2 dh)^2 + 4(1 - hG_2)^{-d} \epsilon \right) (1 - e^{-\lambda h/3}). \tag{48}$$

Iterating the recursive bound,

$$D_{\mathrm{KL}}(\mu_{Nh} \| \pi) \leq \exp(-\frac{\lambda Nh}{3}) D_{\mathrm{KL}}(\mu_0 \| \pi) + 3 \left( 3(M_2 dh)^2 + 4(1 - hG_2)^{-d} \epsilon \right) \lambda^{-1}. \tag{49}$$

$\square$

To further interpret our results with GWG under $W_p$ metric, we assume that the target distribution satisfies *modified log-Sobolev inequality*, which has been considered in many classical works (Adamczak et al., 2017; Bobkov & Ledoux, 2000; Barthe & Roberto, 2008).

**Definition D.1** ( *modified log-Sobolev inequality*)**.** For $q > 1$, we say $\pi$ satisfies the *modified log-Sobolev inequality* mLSI$(q, \lambda_q)$ if the following holds:

$$\mathrm{Ent}_\pi(|f|^q) \leq \frac{q^{q-1}}{\lambda_q} \mathbb{E}_\pi[\|\nabla f\|_q^q], \text{ for all smooth } f: \mathbb{R}^d \to \mathbb{R}.$$

Note that mLSI$(2, \lambda_2)$ reduces to the conventional *log-Sobolev inequality* with constant $\lambda_2$. As a direct corollary of this inequality, for any distribution $\mu$, we take $f = (\frac{\mu}{\pi})^{1/q}$ and thus

$$D_{\mathrm{KL}}(\mu \| \pi) \leq \frac{1}{q\lambda_q} \mathbb{E}_\mu \left\| \nabla \log \frac{\pi}{\mu} \right\|_q^q. \tag{50}$$

**Theorem D.12.** *If we additionally assume that $\pi$ satisfies (50), then under the same conditions of Theorem D.9 with $\varepsilon_k \leq \epsilon$, it holds that*

$$D_{\mathrm{KL}}(\mu_{Nh} \| \pi) \leq \exp(-\frac{q\lambda_q Nh}{12}) D_{\mathrm{KL}}(\mu_0 \| \pi) + 12 \left( A_1 (M_p dh)^q + A_2 (1 - hG_2)^{-d} \epsilon \right) (q\lambda_q)^{-1}.$$

*In particular, if $\epsilon \lesssim \min \left\{ (\frac{M_p}{G_2})^q, (\frac{dM_p}{qG_2})^q \right\}$, we take $h \asymp \frac{\epsilon^{1/q}}{M_p d}$ and then we obtain the guarantee $D_{\mathrm{KL}}(\mu_{Nh} \| \pi) \lesssim \lambda_q^{-1} \epsilon$ after*

$$N = \tilde{\mathcal{O}} \left( \frac{M_p d}{\lambda_q \epsilon^{1/q}} \log \frac{\lambda_q D_{\mathrm{KL}}(\mu_0 \| \pi)}{\epsilon} \right) \qquad \text{iterations.}$$

*Here $\tilde{\mathcal{O}}(\cdot)$ hides all the constant factors that only depend on q.*

*Proof.* For $t \in (kh, (k+1)h)$, we apply Lemma D.7 and (50) and thus

$$\partial_t D_{\mathrm{KL}}(\mu_t \| \pi) \leq -\frac{q\lambda_q}{12} D_{\mathrm{KL}}(\mu_t \| \pi) + A_1 (M_p dh)^q + A_2 (1 - hG_2)^{-d} \epsilon. \tag{51}$$

By Gronwall's inequality,

$$D_{\mathrm{KL}}(\mu_{(k+1)h} \| \pi) \leq e^{-q\lambda_q h/12} D_{\mathrm{KL}}(\mu_{kh} \| \pi) + 12(q\lambda_q)^{-1} \left( A_1 (M_p dh)^q + A_2 (1 - hG_2)^{-d} \epsilon \right) (1 - e^{-q\lambda_q h/12}). \tag{52}$$

Iterating the recursive bound,

$$D_{\mathrm{KL}}(\mu_{Nh} \| \pi) \leq \exp(-\frac{q\lambda_q Nh}{12}) D_{\mathrm{KL}}(\mu_0 \| \pi) + 12 \left( A_1 (M_p dh)^q + A_2 (1 - hG_2)^{-d} \epsilon \right) (q\lambda_q)^{-1}. \tag{53}$$

$\square$

**Remark 2.** *mLSI($q, \lambda_q$) cannot hold for $q > 2$ as mentioned in Barthe & Roberto (2008). However, we only need (50) to hold for all $\mu = \mu_t$ to prove Theorem D.12. Plus, Gentil et al. (2005, 2007) replace $\|\cdot\|_q^q$ with $\max\{\|\cdot\|_2^2, \|\cdot\|_q^q\}$ and show that mLSI can hold for a class of distributions in this way. We leave this for future work.*

**Remark 3.** *Theorem D.12 illustrates how the choice of $q$ will influence the convergence rate of ParVI. On one hand, larger $q$ would reduce the complexity dependence on $\epsilon$. On the other hand, it is generally difficult to predict how $\lambda_q$ will change with $q$. Besides, large $q$ would also increase the difficulty to train the neural net and obtain a well-estimated direction. Overall, it is challenging to determine the optimal $q$ and thus our adaptive method can present significant advantages.*

## D.5 Technical Lemmas

**Lemma D.13.** *For any two matrices $A, B \in \mathbb{R}^{d \times d}$ with positive eigenvalues, the following holds:*

$$|\log \det A - \log \det B| \le d\|A - B\|_2 \max\{\|A^{-1}\|_2, \|B^{-1}\|_2\}$$

*Proof.* Suppose that the eigenvalues of real matrix $(A - B)B^{-1}$ are $\lambda_1, \overline{\lambda}_1, \cdots, \lambda_k, \overline{\lambda}_k \in \mathbb{C}, \lambda_{2k+1}, \cdots \lambda_d \in \mathbb{R}$. Here $\overline{\lambda}_j$ is the complex conjugate of $\lambda_j$. Then it holds that:

$$\log \det A - \log \det B = \log \det(I + (A - B)B^{-1})$$

$$= \log \prod_{j=1}^{d}(1 + \lambda_j)$$

$$\le \sum_{j=1}^{k} \log(1 + \lambda_j)(1 + \overline{\lambda}_j) + \sum_{j=2k+1}^{d} \log(1 + |\lambda_j|)$$

$$\le \sum_{j=1}^{d} \log(1 + |\lambda_j|)$$

$$\le d\|(A - B)B^{-1}\|_2$$

Similarly, we have $\log \det B - \log \det A \le d\|(B - A)A^{-1}\|_2$ and thus we finish the proof. $\square$

**Lemma D.14.** *Define non-negative constants $c_1, c_2$ as:*

$$(c_1, c_2) = \begin{cases} (0, \ 2^{p-q}) & \text{if } q \le 2, \\ \left(3^{-p}, \ \min\left\{3^{q-p}(\frac{p}{q})^{\frac{q-p}{q-1}}(1 - \frac{p}{q})^{\frac{q-p}{p}}, (q-1)^p \left(\frac{(\frac{4}{3})^{\frac{1}{q-1}}}{(\frac{4}{3})^{\frac{1}{q-1}} - 1}\right)^{q-p}\right\}\right) & \text{otherwise.} \end{cases}$$

*Then for any $a, b \in \mathbb{R}$, the following inequality holds:*

$$\left|sgn(a)|a|^{q-1} - sgn(b)|b|^{q-1}\right|^p \le c_1|a|^q + c_2|a - b|^q$$

*Proof.* We shall prove each of the two cases separately.

(1) **Case $q \le 2$.**

If $a, b$ have the same sign, we assume they are positive without loss of generality. Then $|a^{q-1} - b^{q-1}| \le |a - b|^{q-1}$, which implies $|a^{q-1} - b^{q-1}|^p \le |a - b|^q$.

If $a, b$ have different signs, we assume $a \ge 0, b < 0$ without loss of generality. Then by Hölder inequality $a^{q-1} + (-b)^{q-1} \le 2^{2-q}|a - b|^{q-1}$, *i.e.*, $|a^{q-1} - (-b)^{q-1}|^p \le 2^{p-q}|a - b|^q$.

(2) **Case $q > 2$.**

If $a, b$ have different signs, we assume $a \ge 0, b < 0$ without loss of generality. Then $|a^{q-1} + (-b)^{q-1}| \le |a - b|^{q-1}$, which implies $|a^{q-1} - b^{q-1}|^p \le |a - b|^q$.

If $a, b$ have the same sign, we assume they are positive without loss of generality. Note that this inequality is homogeneous, we can let $a = 1$ so that we only need to show for any $b > 0$,

$$\left|1 - b^{q-1}\right|^p \le c_1 + c_2|1 - b|^q. \tag{54}$$

If $b \le 1$, then by simple calculus,

$$
\begin{aligned}
(1 - b^{q-1})^p - c_2(1 - b)^q &\le [(q-1)(1-b)]^p - c_2(1-b)^q \\
&\le c_2^{-\frac{p}{q-p}}(q-1)^{\frac{q+p}{q-p}}(\frac{p}{q})^{\frac{p}{q-p}}(1 - \frac{p}{q}) \\
&\le c_1.
\end{aligned}
$$

If $1 < b \le \left(1 + c_1^{1/p}\right)^{\frac{1}{q-1}} = (\frac{4}{3})^{\frac{1}{q-1}}$, then (54) is trivial.

If $b > (\frac{4}{3})^{\frac{1}{q-1}}$,

$$
\begin{aligned}
(b^{q-1} - 1)^p - c_2(b-1)^q &= (b-1)^p[(\frac{b^{q-1} - 1}{b - 1})^p - c_2(b-1)^{q-p}] \\
&\le (b-1)^p \left([(q-1)b^{q-2}]^p - c_2(b-1)^{q-p}\right) \\
&\le 0.
\end{aligned}
$$

The last inequality is due to $c_2 \ge (q-1)^p \left(\frac{(\frac{4}{3})^{\frac{1}{q-1}}}{(\frac{4}{3})^{\frac{1}{q-1}} - 1}\right)^{q-p} \ge (q-1)^p \left(\frac{b}{b-1}\right)^{q-p}$. $\qquad\square$

## E  Proof of Proposition 5

*Proof.* Note that $g(\cdot) = \frac{1}{p}\| \cdot \|_p^p$ and thus $\nabla g^*(\cdot) = |\cdot|^{q-1} \odot sgn(\cdot)$. By Young's inequality,

$$
\begin{aligned}
\partial_t D_{\mathrm{KL}}(\mu_t \| \pi) = &-\mathbb{E}_{\mu_t}\left\langle \nabla \log \frac{\pi}{\mu_t}, \nabla g^*(\nabla \log \frac{\pi}{\mu_t})\right\rangle \\
&+ \mathbb{E}_{\mu_t}\left\langle \nabla \log \frac{\pi}{\mu_t}, \nabla g^*(\nabla \log \frac{\pi}{\mu_t}) - f_t\right\rangle \\
\le &-(1 - \frac{1}{q}\lambda_1^q)\mathbb{E}_{\mu_t}\left\|\nabla \log \frac{\pi}{\mu_t}\right\|_q^q \\
&+ \frac{1}{p}\lambda_1^{-p}\mathbb{E}_{\mu_t}\left\|\nabla g^*(\nabla \log \frac{\pi}{\mu_t}) - f_t\right\|_p^p.
\end{aligned} \tag{55}
$$

where $\lambda_1$ could by any positive scalar. Here we set $\lambda_1 = 1$ and finish the proof. $\qquad\square$

**Remark 4.** *(21) is not the only way to adaptively choose optimal $p$. In fact, (55) provides a wide range of methods based on different choices of $\lambda_1$ and thus we can obtain different objectives for $p$. We leave it for future work.*

## F  Additional Details of Experiments

### F.1  Gaussian Mixture

We follow the same setting as Dong et al. (2023). The marginal probability of each cluster is 1/10. The number of particles is 1000. For $L_2$-GF, PFG and Ada-GWG, we parameterize $f_w$ as 3-layer neural networks with *tanh* activation function. Each hidden layer has 32 neurons. The inner loop iteration is 5 and we use SGD optimizer with Nesterov momentum (momentum 0.9) to train $f_w$ with learning rate $\eta$=1e-3. The particle step size is 0.1.

For PFG, following Dong et al. (2023), we set the preconditioning matrix $H = \hat{H}^\alpha$, where $\hat{H}$ is the inverse of diagonal variance of particles and $\alpha$ is 1.0.

For Ada-GWG, we set the initial exponent $p_0 = 2$ and learning rate $\tilde{\eta} = 2.5e\text{-}7$.

Figure 5 shows some quantitative comparisons between different algorithms. Here Exp-GF represents GWG with $g(\cdot) = \exp(\|\cdot\|_2^2/(2\sigma^2)) - 1$. The results are the averaged after 10 random trials. We can observe that Ada-GWG can obtain highly-accurated samples within fewer iterations.

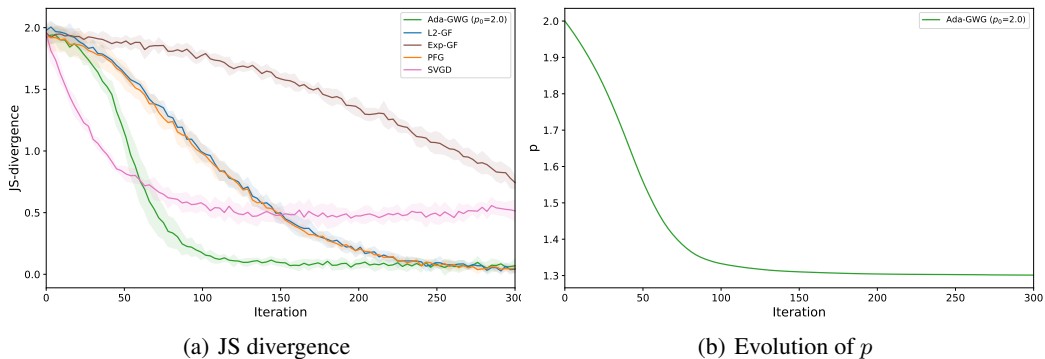

      (a) JS divergence                   (b) Evolution of $p$

Figure 5: Quantitative results in Gaussian Mixture experiment

### F.2 Monomial Gamma

On heavy tailed distributions, the number of particles is 1000. For GWG and Ada-GWG, the neural network structure is the same as which in the Gaussian mixture experiment. The inner loop iteration is also 5, but we use Adam optimizer with learning rate $\eta$=1e-3 to train $f_w$ for better stability. The particle step size is 1e-3.

For Ada-GWG, we set the initial exponent $p_0 \in \{1.5, 2.0, 2.2\}$ and learning rate $\tilde{\eta} = 1$.

We run the experiment on 4 random seeds. The average results and the variances are represented in the figure using lines and shades.

### F.3 Conditioned Diffusion

The procedure to generate the true path is exactly the same as in Detommaso et al. (2018). For PFG and Ada-GWG, we parameterize $f_w$ as 3-layer neural networks with *tanh* nonlinearities. Each hidden layer has 200 neurons. The inner loop $N'$ is selected from $\{1, 5, 10, 15\}$ to get the best performance. $f_w$ is pre-trained for 100 iterations before particle updates and we use Adam optimizer with learning rate $\eta$=1e-3 to train $f_w$. The particle step size is $3e-3$ for Ada-GWG and PFG.

For Ada-GWG, we set the initial exponent $p_0 = 2.2$ and learning rate $\tilde{\eta} = 0.001$. The gradient of $A(p)$ is also clipped within $[-0.1, 0.1]$.

For PFG, we set the preconditioning matrix $H = \hat{H}^\alpha$, where $\hat{H}$ is the inverse of diagonal variance of particles and $\alpha$ is chosen from $\{0.1, 0.5, 1.0\}$ to obtain the best performance.

For SVGD, we use RBF kernel $\exp(-\dfrac{\|x-y\|^2}{h})$ where $h$ is the heuristic bandwidth (Liu & Wang, 2016). The initial step size is 1e-3 and is adjusted by AdaGrad.

Additionally, we run LMC with step size 1e-4 for 10000 iterations as the ground truth for posterior distribution.

### F.4 Bayesian Neural Networks

Our experiment settings are almost similar to SVGD (Liu & Wang, 2016). For the UCI datasets, the datasets are randomly partitioned into 90% for training and 10% for testing. Then, we further split

the training dataset by 10% to create a validation set for hyperparameter selection as done in (Liu & Wang, 2016). For $L_2$-GF and Ada-GWG, we parameterize $f_w$ as 3-layer neural networks. Each hidden layer has 300 neurons, and we use LeakyReLU as the activation function with a negative slope of 0.1. The inner loop $N'$ is selected from $\{1, 5, 10\}$. We use the Adam optimizer and choose the learning rate from $\{0.001, 0.0001\}$ to train $f_w$.

For Ada-GWG, we choose the initial exponent $p_0$ from $\{3, 4\}$ and set the learning rate $\tilde{\eta} = 0.0001$. The gradient of $A(p)$ is clipped within [-0.2, 0.2]. We select the step size of particle updates from $\{0.0001, 0.0002, 0.0005, 0.001\}$. For SVGD, we use the RBF kernel as done in (Liu & Wang, 2016). For SVGD, $L_2$-GF, and Ada-GWG, the iteration number is chosen from $\{2000, 4000\}$ to converge. For SGLD, the iteration number is set to 10000 to converge.

## G  Limitations and Future Work

**Estimating Wasserstein gradient by neural networks.**   Our formulation leverages the capability of neural networks to estimate the generalized Wasserstein gradient. This approach successfully resolves the problem of kernel design for conventional ParVI methods. However, in high dimensional regime, the design of neural network structure is still important but subtle. Besides, the computation cost is also expensive. We expect more efficient algorithms on training neural works to approximate Wasserstein gradient, *e.g.* Wang et al. (2022).

**Better adaptive method.**   Our Ada-GWG method is based on the idea of maximizing the decent rate of KL divergence and heavily relies on an accurate estimation of generalized Wasserstein gradient. We update exponent $p$ by simply gradient ascent which may cause severe numerical instability. Although this can be alleviated by clipping, it is still delicate when the target distribution is complex.

**General Young function class.**   In this paper, we only consider the function class $\left\{ \frac{1}{p} \| \cdot \|_p^p : p > 1 \right\}$, which is still limited. We expect a more general function class that both have numerical stability and can perfectly capture the information from score function. The characteristics of Young function class may be instructive. How to design an adaptive algorithm on a more general class is also challenging and important. We leave this to future research.

**Shortness of theoretical analysis.**   Although we provide convergence guarantee under weak assumptions, our analysis is still preliminary and we believe these results can be strengthened. There are also other important extensions to consider. For example, our analysis is based on the population loss, which is an asymptotic result based on infinite particles limit. We believe this framework can be also generalized to finite-particle system like SVGD (Korba et al., 2020). Moreover, we only consider Young functions that have the form of $\| \cdot \|_p^p$ or are strongly convex and strongly smooth. We believe that better-designed Young functions may have more advantageous theoretical properties, *e.g.*, Wasserstein Newton flow (Wang & Li, 2020).

