# OpenReview forum: "Particle-based Variational Inference with Generalized Wasserstein Gradient Flow"
_NeurIPS.cc/2023/Conference — NeurIPS 2023 poster_

### Official Review · Reviewer_Yghj · 2023-06-18

**Soundness:** 4 excellent
**Presentation:** 3 good
**Contribution:** 4 excellent
**Rating:** 7
**Confidence:** 3

**Summary:**

This paper proposes to compute gradient flows of the KL divergence in the space of probability measures endowed by a generalization of the Wasserstein distance, which uses more general cost based on Young functions, and called Generalized Wasserstein gradient flows (GWGF).

Authors provide the forward Euler scheme of such gradient flow and observe that the choice of the Young function has an impact on the convergence rate of the flow. Thus, besides proposing an algorithm to solve the GWGF with neural networks and analyzing its convergence rate, they also propose an algorithm to adapt the Young function in order to have accelerated convergence rate.

Finally, they propose several applications and compare with other gradient flows based methods such as the SVGD or the preconditioned functional gradient flow. They show that they obtain consistently better results and also illustrate the benefits of the adaptative algorithm.


**Strengths:**

The paper is overall pretty clear and the idea to perform gradient flows in this class of OT problems with more general costs is very appealing.

- Interesting idea well motivated
- Good Theoretical analysis
- Many applications with comparisons with other methods

**Weaknesses:**

- It is not a big weakness, but the class of Young functions investigated seems not very big (as it only experimentally considers $c'x,y)=\|x-y\|_p^p$ with $p\in [1.1,4]$). A test with a less common cost could have been interesting.

**Questions:**

The existence of the OT map for the Wasserstein cost is not discussed and is assumed. In [1] (Section 1.3), the result is stated when the cost is of the form $c(x,y)=g(x-y)$ with $g$ strictly convex. In the Young function class, $g$ is only assumed to be convex. So I think I am missing something here.

In equation (7), I believe that a $\frac{1}{2h}$ is missing under the Wasserstein cost.

In Figure 1, the SVGD is not reported. Is it because it is hard to find good hyperparameters which converge for multimodal distributions? I think it would have been nice to add a plot of the evolution of p in the adaptative version of GWG.

Equation (13) is referred to as the Euler-Maruyama scheme. In my opinion, this is the forward-euler discretization and not the Euler-Maruyama scheme which refers to the discretization of SDEs.

[1] Santambrogio, F. (2015). Optimal transport for applied mathematicians. Birkäuser, NY, 55(58-63), 94.

**Limitations:**

The limitations are addressed in appendix.

---

> ### Author Rebuttal · Authors · 2023-08-09
>
> Thank you for your careful review and valuable questions! We address your comments and questions as below.
>
> ### Weaknesses
> `W1`: The class of Young functions investigated seems not very big.
>
> A1: The choices of Young functions are quite delicate since it should strick a balance between accelerating convergence and maintaining numerical stability. Apart from the $\\|\cdot\\|_p^p$ and the preconditioned quadratic form proposed in ([1]), we further test other functions like MGF $g(\\cdot)=\exp\left(\\|\cdot\\|_2^2/(2\\sigma^2)\right)-1$ on the 2D-Gaussian mixture model. The results show that this class of Young functions may suffer from numerical instability and poor acceleration performance, and thus is not preferred. For even more general $g$ functions, one interesting future direction is that we can parameterize $g$ as an ICNN [2].
>
> ### Questions
> `Q1`: The existence of the OT map for the Wasserstein cost is not discussed and is assumed.
>
> A1: Thanks for your advice! The general convexity is indeed not sufficient to ensure the existence of OT map. To be mathematically rigorous, we will add the "strictly convex" requirement in our revision.
>
> `Q2`: In equation (7), I believe that a $\\frac{1}{2h}$ is missing under the Wasserstein cost.
>
> A2: We beg to differ. Actually, our definition of cost function is $c_h(x,y):= g(\\frac{x-y}{h})h$ in Theorem 1. For sanity check, if $g(\cdot)=\frac{1}{2}\\|\cdot\\|^2$, then in eq (7), $W\_{c\_h}(\\mu, \\mu\_{kh})=\\frac{1}{2h}W\_2^2(\\mu, \\mu\_{kh})$. This corresponds to the $L\_2-GF$ ([3]).
>
> `Q3`: In Figure 1, the SVGD is not reported. It would have been nice to add a plot of the evolution of p in the adaptative version of GWG.
>
> A3: We reported the results of SVGD and the plot of the evolution of $p$ in the rebuttal PDF. Although SVGD may be fast at first, it fails to converge to target distribution in 300 iterations. On the other hand, after 50 iterations, the neural network in Ada-GWG can estimate the vector field accurately and exhibits significant acceleration effect.
>
>
> `Q4`: Equation (13) should be referred to as the forward-Euler discretization.
>
> A4: Thanks for your advice, we will correct this mistake in our revision.
>
> [1] Dong, Hanze, et al. ‘Particle-Based Variational Inference with Preconditioned Functional Gradient Flow’. arXiv Preprint arXiv:2211. 13954, 2022.
>
> [2] Amos, Brandon, et al. ‘Input Convex Neural Networks’. International Conference on Machine Learning, PMLR, 2017, pp. 146–155.
>
> [3] Ambrosio, Luigi, et al. Gradient Flows: In Metric Spaces and in the Space of Probability Measures. Springer Science & Business Media, 2005.

---

> > ### Comment · Reviewer_Yghj · 2023-08-14
> >
> > Thank you for the response, the clarifications and the additional experiments. I am overall convinced by this work and will keep my rating unchanged.

---

### Official Review · Reviewer_GLuk · 2023-06-19

**Soundness:** 3 good
**Presentation:** 3 good
**Contribution:** 3 good
**Rating:** 6
**Confidence:** 5

**Summary:**

The paper's primary focus is the use of a generalized Wasserstein gradient flow of the KL divergence for solving the sampling problem in a particle-based variational inference framework, which they name Generalized Wasserstein Gradient Descent (GWG). Unlike the usual Wasserstein gradient flow of the KL, which corresponds to the Langevin diffusion, the implementation of this generalized gradient flow is more complex, hence the authors propose learning the gradient vector field via a neural network. They further back their proposal with a theoretical analysis which indicates that if the neural network learns the vector fields sufficiently well, guarantees can be obtained in a $L^q$ variant of the Fisher information.

**Strengths:**

The paper presents a novel sampling algorithm using generalized Wasserstein gradient flows, which provides an interesting avenue for designing alternative approaches to SVGD.

The algorithm is backed by theory. The authors present robust guarantees about the convergence of the proposed method given certain conditions on the learning ability of the neural network.


**Weaknesses:**

While the approach presented in the paper is interesting, the technical novelty appears to be limited, as the underlying analysis follows the framework presented by Balasubramanian et al.

Assumptions 1 and 2, which concern the neural network's capability to accurately learn vector fields, form a significant portion of the work's theoretical underpinning. However, the realism of these assumptions in practical scenarios is unclear.


**Questions:**

In the context of existing literature, the reference to Bernton (2018) on line 88 seems to be out of place. Would it not be more accurate to reference the Jordan–Kinderlehrer--Otto (JKO) paper?

Lines 170-171 contain an inaccurate claim regarding the dissipativity assumption in Balasubramanian et al. (the dissipativity assumption they use is much weaker than the log-Sobolev inequality) – could this be corrected?

The paper heavily relies on Assumptions 1 and 2 regarding the accuracy of the neural network's learning of the vector fields. Could the authors shed more light on how realistic these assumptions are and discuss any empirical evidence that might support these assumptions?


**Limitations:**

The fact that the analysis hinges on an assumption for which it is unclear if it holds in practice, namely accurate estimation of the vector field via neural networks, should be addressed more clearly.

---

> ### Author Rebuttal · Authors · 2023-08-09
>
> Thank you for your careful review and valuable questions! We address your comments and questions as below.
>
> #### Weaknesses
>
> `W1`: The technical novelty appears to be limited, as the underlying analysis follows the framework presented by Balasubramanian et al.
>
> A1: We indeed borrow some ideas from [1]. However, our analysis is much more involved and exhibits great novelty for analyzing particle-based VI. As described in the beginning of Appendix D, we need to bound both discretization error and estimation error.
>   1. For discretization error, the main technical difficulties include how to ensure the smoothness of particle distribution. Note that we even do not assume the smoothness of target distribution.
>   2. For estimation error, since neural networks can only give an estimation of the optimal vector field, the difficulty lies in how to control the evolution trajectory of particle distribution with inaccurate vector field.
>
> Most importantly, the two error terms are actually entangled with each other and the bound of one term will depend on the other. Therefore, the analysis of both terms needs to be handled with great care to achieve the SOTA rate.
>
> To the best of our knowledge, our result is the first non-asymptotic analysis of particle-based VI with functional gradient. And it's also for the first time, that one can show that particle-based VI is able to outperform traditional Langevin Monte Carlo theoretically. We believe that our techniques can motivate further theoretical analysis of particle-based VI methods.
>
>
>
> #### Questions
>
> `Q1`: In the context of existing literature, the reference to Bernton (2018) on line 88 seems to be out of place.
>
> A1: Thanks for your advice! We will correct the reference in our revision.
>
> `Q2`: Lines 170-171 contain an inaccurate claim regarding the dissipativity assumption in Balasubramanian et al.
>
> A2: Thanks for your advice! We will correct it in our revision. The assumption made by Balasubramanian et al.[1] is to control the order of growth of the potential function, which is more general than the common dissipativity assumption. This assumption indeed does not imply log-Sobolev inequality. However, it is still quite strong and our analysis does not rely on it. Therefore, our claim still holds that "particle-based methods can outperform the traditional LMC as long as the neural nets can estimate the vector field accurately".
>
> `Q3`: Could the authors shed more light on how realistic Assumptions 1 and 2 are and discuss any empirical evidence that might support these assumptions?
>
> A3: We discuss the Assumptions 1 and 2 in two aspects.
> 1. **Theoretical justification for Assumption 1 and 2**
> Given current particle distribution $\mu$, since we estimate the vector field by maximizing eq (14), we can define the training loss $\\mathcal{L}\_{\\text{train}}(v):=\\mathbb{E}\_{\\mu} [\\langle \\nabla\\log\\frac{\\pi}{\\mu}, v\\rangle - g(v)] $. The maximizer is $v^*=\\nabla g^*(\\nabla\\log\\frac{\\pi}{\\mu})$ and the maximum value is $\\mathcal{L}\_{\\text{train}}^*:=\\mathcal{L}\_{\\text{train}}(v^*)<\\infty$. Similar to Lemma D.14, we can show that for any $p>1$ and any arbitrarily small $\\varepsilon\_1>0$, if $g(\\cdot)=\frac{1}{p}\\|\cdot\\|_p^p$, there exists $\\varepsilon_2:=\\varepsilon_2(\\varepsilon\_1, p)>0$, such that
>     $$
>         \\mathbb{E}\_{\\mu} [\\|v-\nabla g^*(\\nabla\\log\\frac{\\pi}{\mu})\\|_p^p] \\leq \\varepsilon\_1 \\mathcal{L}\_{\\text{train}}^* + \\varepsilon\_2 [\\mathcal{L}\_{\\text{train}}^*-\\mathcal{L}\_{\\text{train}}(v)].
>     $$
>
>     This shows that Assumption 1 is realistic as long as we can optimize the training loss function well (Assumption 2 is similar). We also discuss the asymptotic rate of estimating optimal $v^*$ with finite particles in Appendix C to further justify our Assumption 1 and Assumption 2.
>
> 2. **Empirical evidence**
>     Since the score of particle distribution is unknown, we cannot compute the objective in Assumption 1 or 2 directly. However, given the great empirical performance of score-based generative model, we believe that the capability of modern neural networks is sufficient to estimate the optimal vector field.
>
> [1] Balasubramanian, Krishnakumar et al. “Towards a Theory of Non-Log-Concave Sampling: First-Order Stationarity Guarantees for Langevin Monte Carlo.” Annual Conference Computational Learning Theory (2022).

---

> > ### Comment · Reviewer_GLuk · 2023-08-13
> >
> > Thank you for your response.
> >
> > Regarding smoothness, although you do not assume smoothness of the target, you assume the existence of accurate approximations of the true vector field via vector fields which are smooth, which is morally the same kind of assumption but less transparent. Therefore, I do not think this particularly adds to the novelty of the analysis.
> >
> > In general, I think you are over-claiming. In your rebuttal, you write that this is the first time that particle VI is theoretically shown to outperform Langevin Monte Carlo, but this paper relies on far more assumptions than LMC which cannot be checked. I think the proposed algorithm is interesting and the theory does lend some credence to it, but please avoid misleading comparisons. The analysis of LMC relies on very few, well-established and checkable assumptions, and in particular does not require guarantees for training a neural network (which is practically impossible at present) and infinitely many particles.
> >
> > I will keep my current score.

---

> > > ### Author Response · Authors · 2023-08-13
> > >
> > > Thanks for your suggestions! Our results indeed depend on some assumptions which cannot be checked explicitly and we acknowledge that our claim may be overly assertive. We will avoid such strong claims in our revision.
> > >
> > > Furthermore, it should be clarified that to remove the smoothness assumption of target distribution is not the main contribution of this work. Instead,  our advantage is that we are able to eliminate the additional growth condition in Balasubramanian et al. We hope that our theory part can motivate further research on particle-based VI.

---

### Official Review · Reviewer_QMGR · 2023-07-05

**Soundness:** 3 good
**Presentation:** 2 fair
**Contribution:** 3 good
**Rating:** 5
**Confidence:** 5

**Summary:**

This paper proposes a general version to solve Wasserstein Gradient Flow for Particle-based VI. The authors show that this approach offers strong convergence guarantees and better performance over SVGD and other methods, as evidenced by extensive experiments on both simulated and real data sets. The paper also introduces a novel theoretical convergence guarantee of ParVIs with neural-net-estimated vector fields for generalized Wasserstein gradient flow.


**Strengths:**

GWG is an interesting extension of ParVI, which extend the quadratic distance to more general metrics.

Both theoretical and empirical justification suggest the improvement of the proposed algorithm.


**Weaknesses:**

The title Particle-based Variational Inference with Generalized Wasserstein Gradient Flow is too close to (Dong et al. 2023). I would suggest to use a new title to highlight the improvements.

More intuitive illustration of the method would be more helpful.

The implementation details and codes are missing.


**Questions:**

What is a proper heuristic strategy to choose $p$?

Is there any geometric interpretation of the proposed $g$?



**Limitations:**

The paper lacks some intuitive demonstration of the propose $g$, which would be very helpful for readers.

---

> ### Author Rebuttal · Authors · 2023-08-09
>
> Thank you for your careful review and valuable questions! We address your comments and questions as below.
>
> #### Weaknesses
> `W1`: The title Particle-based Variational Inference with Generalized Wasserstein Gradient Flow is too close to (Dong et al. 2023).
>
> A1: Thanks for the suggestion! We will modify our title in our revision.
>
> `W2`: More intuitive illustration of the method.
>
> A2: As mentioned in Section 3.2, our motivation and intuition for GWG is to accelerate particle-based VI in terms of faster decay of KL divergence, which is determined by equation (10). We showed in Example 1 that conventional $L_2$-GF may suffer from slow decaying rate of KL. In this sense, it is natural to generalize the formulation of minimizing movement scheme and also the corresponding training method. We will make this points clearer in our revision.
>
> `W3`: The implementation details and codes are missing.
>
> A3: We include all the implementation details in Appendix F of the supplementary material. We are still cleaning up the code and it will be released soon.
>
>
> #### Questions
> `Q1`: What is a proper heuristic strategy to choose $p$?
>
> A1: As claimed in Section 3.2, our primary goal is to achieve faster decay of KL divergence. Hence the criterion to choose $p$ is to enlarge the magnitude of derivatives of KL, i.e. $\\mathbb{E}\_{\\mu\_t} \\| \nabla\\log\\frac{\\pi}{\\mu_t}\\|_q^q$, where $q=p/(p-1)$. Informally, $p$ should be small (so that $q$ is large) if $\\|\nabla\\log\\pi - \\nabla\\log\\mu_t \\|$ is large and vise versa. The intuition is to apply a larger penalty term when there is a significant difference between the target score and the particle score.
> In practice, this heuristic strategy is difficult to carry out. Therefore, based on this criterion, we propose Ada-GWG to choose $p$ automatically by increasing an lower bound of $|\\partial\_t{\\mathrm{KL}}|$.
>
> `Q2`: Is there any geometric interpretation of the proposed $g$?
>
> A2: Different $g$ corresponds to different Wasserstein metric and Wasserstein space. If $g(\cdot)=g_0(\\|\cdot\\|)$, where $\\|\cdot\\|$ can be any norm in the Euclidean space and $g_0$ satisfies some mild assumptions, we can easily show by generalized Minkowski's inequality ([1]) that $g_0^{-1}(W\_{c\_h}(\\mu,\\nu))$ is a well-defined metric and hence $\mathcal{P}_{c_h}$ defined in Theorem 1 is a Wasserstein space. The general class of functions for $g$ allows us to explore the underlying structure of different probability spaces and further utilize this geometric structure to accelerate convergence.
>
> [1] Mulholland, H. P. ‘On Generalizations of Minkowski’s Inequality in the Form of a Triangle Inequality’. Proceedings of The London Mathematical Society, 1949, pp. 294–307.

---

> > ### Comment · Reviewer_QMGR · 2023-08-14
> >
> > Thanks for your response and it help me to understand the paper better. I would suggest the authors add more practical interpretations and justifications in future versions.

---

> > > ### Author Response · Authors · 2023-08-16
> > >
> > > Thanks for your suggestion! We will modify our paper accordingly in our revision.

---

### Official Review · Reviewer_t31F · 2023-07-06

**Soundness:** 3 good
**Presentation:** 3 good
**Contribution:** 2 fair
**Rating:** 4
**Confidence:** 3

**Summary:**

This paper proposes a novel particle-based variational inference framework based on generalized Wasserstein gradient flow of KL divergence, named generalized Wasserstein gradient descent (GWG).  The strong convergence guarantee of the proposed algorithm is provided. The authors also provide an adaptive version based on Wasserstein metric to accelerate convergence.


**Strengths:**

- The paper is well-written and clearly-organized.
- It is quite novel to replace the standard Wasserstein-2 metric by using a Wasserstein metric with cost function defined by Young function.
- Numerical results support effectiveness of proposed method.


**Weaknesses:**

- The description of main results (Theorem 2 and Theorem 3) are unclear. It is not specified how the vector-valued function vk is learned. Is vk the maximizer of equ (14) with finite samples?
- The convergence result only holds for \bar \mu_{Nh}, which is the average of path instead of the end-iteration solution \mu_{Nh}.
- The numerical results do not show a quanitative convergence improvement of the proposed Ada-GWG with other methods like SVGD, L2-GF or PFG. For instance, It would be better to compare these method in the form of Figure 2, while you plot KL divergence or even maximum mean discrepancy between current iteration and posterior as a function of iteration number or cpu time.

Typos:
- In Algorithm 1, it should be \hat A(p_k) instead of \hat A(p).

**Questions:**

- How Assumption 3 on the smoothness of neural networks are ensured in practice? How to estimate the constants G_p and M_p given a neural network?


**Limitations:**

Yes.

---

> ### Author Rebuttal · Authors · 2023-08-09
>
> Thank you for your careful review and valuable questions! We address your comments and questions as below.
>
> ### Weaknesses
> `W1`: The description of main results (Theorem 2 and Theorem 3) are unclear. It is not specified how the vector-valued function $v\_k$ is learned. Is $v\_k$ the maximizer of eq (14) with finite samples?
>
> A1: Exactly! $v\_k$ is the neural net learned by maximizing equation (14) with finite samples. But here for the theoretical part, we only consider the infinite-particle limit, which is very common in the theoretical analysis of particle-based VI (like [1] [2]). To bridge this gap, we discuss the asymptotic normality of the neural net estimator in Appendix C of the supplementary material. But in general, analyzing the convergence of particle-based VI with finite particles is very difficult, and we leave this for future work as mentioned in Appendix G.
>
>
> [1] Liu, Qiang. “Stein Variational Gradient Descent as Gradient Flow.” NIPS (2017).\
> [2] Salim, Adil et al. “A Convergence Theory for SVGD in the Population Limit under Talagrand's Inequality T1.” International Conference on Machine Learning (2021).
>
> `W2`: The convergence result only holds for $\bar{\mu}\_{Nh}$, which is the average of path instead of the end-iteration solution $\mu\_{Nh}$.
>
> A2: (1) In the literature of non-log-concave sampling, it is generally difficult to attain the convergence rate of the end-iteration solution ([1], [2]). If the target distribution satisfies the log-Sobolev inequality, then we are able to get convergence result of $\\mu\_{Nh}$. Please refer to Appendix D.3 of the supplementary material for more discussions. (2) On the other hand, we can still sample from $\\bar{\\mu}\_{Nh}$ by sampling $t\\sim \\mathrm{Unif}[0,Nh]$ first and then getting a sample from $\\mu\_t$ by $X\_t = X\_{kh} + (t-kh)v\_k(X\_{kh})$ where $k=[\\frac{t}{h}]$. Therefore, we don't think this is a serious drawback.
>
> [1] Balasubramanian, Krishnakumar et al. “Towards a Theory of Non-Log-Concave Sampling: First-Order Stationarity Guarantees for Langevin Monte Carlo.” Annual Conference Computational Learning Theory (2022).
> [2] Korba, Anna et al. “A Non-Asymptotic Analysis for Stein Variational Gradient Descent.” ArXiv abs/2006.09797 (2020): n. pag.
>
> `W3`: The numerical results do not show a quantitative convergence improvement of the proposed Ada-GWG with other methods like SVGD, L2-GF or PFG. For instance, It would be better to compare these method in the form of Figure 2, while you plot KL divergence or even maximum mean discrepancy between current iteration and posterior as a function of iteration number or cpu time.
>
> A3: Thanks for your advice! We plotted the Test RMSE of BNN experiments in Appendix F.4 and it shows that even with unproperly selected $p\_0$, Ada-GWG makes significant improvements and demonstrates comparable performance to the optimal choice. We further plot the JS-divergence of Gaussian Mixture experiment in the PDF file of global response and will add them in our revision. The results also indicate that Ada-GWG can accelerate convergence compared with the baselines.
>
>
> `W4`: In Algorithm 1, it should be \hat A(p\_k) instead of \hat A(p).
>
>
> A4: Thanks for catching the typos, and we have corrected these typos in our revision.
>
>
> ### Questions
>
> `Q1`: How Assumption 3 on the smoothness of neural networks are ensured in practice? How to estimate the constants $G\_p$ and $M\_p$ given a neural network?
>
> A1: (1) Since for any $p\_1\geq p\_2>1, \text{and}\ x\in \\mathbb{R}^d\\backslash\\{0\\}$, $d^{1/p\_1-1/p\_2}\\leq\\frac{\\|x\\|\_{p\_1}}{\\|x\\|\_{p\_2}}\leq 1$, the constants $G\_p$ and $M\_p$ differ from $G\_2$ and $M\_2$ by a factor of $d^{|1/2-1/p|}$ at most. Moreover, $G\_2$ is a standard assumption on the smoothness of neural nets in the literature of score matching ([1]). And as mentioned in Lines 155-156, $M\_2$ corresponds to the Lipschitz constant of the Hessian of $\\log\\pi$ informally, which is also a widely-used assumption to analyze Langevin Monte Carlo ([2]). Therefore, we think Assumption 3 is reasonable to assume. In practice, we can utilize a regularizer (e.g. weight decay) to avoid overfitting and control the smoothness if necessary. (2) To estimate $G\_p$ and $M\_p$, we can compute the gradients of neural nets at current particles in each iteration and use the one with largest gradient norm for estimation. We did so in the experiment of conditioned diffusion. The results are shown in the PDF file of global response. The magenitudes of $G\_2$ and $M\_2$ remain in a reasonable range, further justifying our Assumption 3.
>
> [1] Lee, Holden, et al. ‘Convergence for Score-Based Generative Modeling with Polynomial Complexity’. Advances in Neural Information Processing Systems, vol. 35, 2022, pp. 22870–22882.
>
> [2] Mou, Wenlong, et al. ‘Improved Bounds for Discretization of Langevin Diffusions: Near-Optimal Rates without Convexity’. Bernoulli, vol. 28, no. 3, Bernoulli Society for Mathematical Statistics and Probability, 2022, pp. 1577–1601.

---

> > ### Comment · Reviewer_t31F · 2023-08-14
> >
> > Thanks for the response. I'd like to keep my score.

---

> > > ### Author Response · Authors · 2023-08-16
> > >
> > > Thanks for your response. Please feel free to let us know if you have further questions!

---

### Author Rebuttal · Authors · 2023-08-09

We thank all reviewers for their constructive feedback, and will modify our paper accordingly in our revision. We address some of the common issues raised by the reviewers below.

**Justification for Assumption 1 and 2**

1. `Theoretical justification`
 Given current particle distribution $\mu$, since we estimate the vector field by maximizing eq (14), we can define the training loss $\\mathcal{L}\_{\\text{train}}(v):=\\mathbb{E}\_{\\mu} [\\langle \\nabla\\log\\frac{\pi}{\mu}, v\\rangle - g(v)] $. The maximizer is $v^*=\\nabla g^*(\\nabla\\log\\frac{\\pi}{\\mu})$ and the maximum value is $\\mathcal{L}_{\\text{train}}^*:=\\mathcal{L}\_{\\text{train}}(v^*)<\\infty$. Similar to Lemma D.14, we can show that for any $p>1$ and any arbitrarily small $\\varepsilon\_1>0$, if $g(\\cdot)=\\frac{1}{p}\\|\\cdot\\|\_p^p$, there exists $\\varepsilon\_2:=\\varepsilon\_2(\\varepsilon\_1, p)>0$, such that
    $$
        \\mathbb{E}\_{\\mu} [\\|v-\\nabla g^*(\\nabla\\log\\frac{\\pi}{\\mu})\\|\_p^p] \\leq \\varepsilon\_1 \\mathcal{L}\_{\\text{train}}^* + \\varepsilon\_2 [\\mathcal{L}\_{\\text{train}}^*-\\mathcal{L}\_{\\text{train}}(v)].
    $$

    This shows that Assumption 1 is realistic as long as we can optimize the training loss function well (Assumption 2 is similar). We will add this result in our revision. We also discuss the asymptotic rate of estimating optimal $v^*$ with finite particles in Appendix C to further justify our Assumption 1 and Assumption 2.

2. `Empirical evidence`
 Since the score of particle distribution is unknown, we cannot compute the objective in Assumption 1 or 2 directly. However, given the great empirical performance of score-based generative model, we believe that the capability of modern neural networks is sufficient to optimize the training loss and estimate the optimal vector field.


**Theoretical novelty and highlights**

Our analysis is very involved and exhibits great novelty for analyzing particle-based VI. As described in the beginning of Appendix D, we need to bound both discretization error and estimation error.
  1. For discretization error, the main technical difficulties include how to ensure the smoothness of particle distribution. Note that we even do not assume the smoothness of target distribution.
  2. For estimation error, since neural networks can only give an estimation of the optimal vector field, the difficulty lies in how to control the evolution trajectory of particle distribution with inaccurate vector field.

Most importantly, the two error terms are entangled with each other and the bound of one term will depend on the other. Therefore, the analysis of both terms needs to be handled with great care to achieve the SOTA rate.

To the best of our knowledge, our result is `the first non-asymptotic analysis of particle-based VI with functional gradient. And it's also for the first time, that one can show that particle-based VI is able to outperform traditional Langevin Monte Carlo theoretically`. We believe that our techniques can motivate further theoretical analysis of particle-based VI methods.


**Quantitative results of improvement**

We plotted the Test RMSE of BNN experiments in Appendix F.4 (also included in the PDF file of global response). It shows that even with improperly selected $p\_0$, Ada-GWG makes significant improvements and demonstrates comparable performance to the optimal choice. This suggests that Ada-GWG is able to push $p\_0$ towards the optimal $p$. We further plot the JS-divergence of Gaussian Mixture experiment in the PDF file and will add them in our revision. The results also indicate that Ada-GWG can accelerate convergence significantly compared with other baselines.

We hope our response has adequately addressed the reviewers' questions and concerns, and look forward to reading any other additional comments.

---

### Decision · Program_Chairs · 2023-09-21

**Decision:**

Accept (poster)

**Comment:**

Throughout the reviews, the reviewers acknowledge the novelty to develop particle-based variational inference methods by gradient flows under generalized metrics, and a non-asymptotic analysis that also takes the estimation error of neural network required in the method into account (though roughly following a recent proof framework). There are also concerns on the reality of the assumptions, the significance of empirical performance, and some imprecise statements, which seem to have been addressed to some extent in the rebuttal. After a few discussions with reviewers, the overall evaluation tends towards outweighing the contributions. I hence recommend an accept.

Nevertheless, apart from implementing the mentioned amendments in rebuttal into the final paper, the authors also need to restrain from overclaims unless clearly analyzed, for example, how it is "the first non-asymptotic analysis of particle-based VI with functional gradient" compared to e.g., Korba et al. (2020), and what is additionally required to "show that particle-based VI is able to outperform traditional Langevin Monte Carlo theoretically". The authors are also expected to explain more on the formulation for derivation, e.g., how the extension of the MMS is a solid mathematical construction that can be interpreted as a gradient flow, whether the employed Wasserstein loss is indeed a distance as required by MMS, and what is the difference from the standard MMS using the p-Wasserstein distance.